# On the Optimality of the Median-of-Means Estimator under Adversarial Contamination

Xabier de Juan[1]    Santiago Mazuelas[1,2]
[1]Basque Center of Applied Mathematics (BCAM)
[2]IKERBASQUE-Basque Foundation for Science
{xdejuan, smazuelas}@bcamath.org

## Abstract

The Median-of-Means (MoM) is a robust estimator widely used in machine learning that is known to be (minimax) optimal in scenarios where samples are i.i.d. In more grave scenarios, samples are contaminated by an adversary that can inspect and modify the data. Previous work has theoretically shown the suitability of the MoM estimator in certain contaminated settings. However, the (minimax) optimality of MoM and its limitations under adversarial contamination remain unknown beyond the Gaussian case. In this paper, we present upper and lower bounds for the error of MoM under adversarial contamination for multiple classes of distributions. In particular, we show that MoM is (minimax) optimal in the class of distributions with finite variance, as well as in the class of distributions with infinite variance and finite absolute $(1+r)$-th moment. We also provide lower bounds for MoM's error that match the order of the presented upper bounds, and show that MoM is sub-optimal for light-tailed distributions.

## 1  Introduction

The Median-of-Means (MoM) estimator is a widely used one-dimensional estimator of the mean. First introduced in the 1980s in [1], it has recently gained significant attention from the robust machine learning community. For example, MoM has enabled the development of robust alternatives for empirical risk minimization [2–4], kernel methods [5, 6], and clustering techniques [7]. MoM is an attractive robust univariate mean estimator in scenarios affected by heavy tails [8–11]. In particular, MoM is known to be a (minimax) optimal estimator of the mean in the i.i.d. scenario for heavy-tailed distributions [10], together with the trimmed mean [12], Catoni's M-estimator [13], and the Lee-Valiant estimator [14]. Moreover, as shown in [15, 16], MoM is also an adequate estimator in situations where data samples are contaminated by an adversary.

Adversarial contamination is a general type of data attack in which the adversary is allowed to first inspect the clean samples, then remove a proportion of them, and finally add new samples [17]. These types of attacks can be specifically designed to maximize the damage caused to a subsequent learning process [18, 19]. Adversarial contamination can be especially harmful in fields where data integrity is crucial, such as cybersecurity [20], biometrics [21], and autonomous driving [22]. Motivated by these concerns, the robust machine learning community has shown strong interest in the problem of mean estimation under adversarial contamination [17, 23–25].

The (minimax) optimal error under adversarial contamination depends on both the fraction of contaminated samples $\alpha$ and the class of distributions considered [24]. The optimal error exhibits two regimes: (1) for a reduced enough number of samples, the optimal error matches that in the i.i.d. scenario; (2) as the number of samples increases, the optimal error plateaus at a level determined by $\alpha$ and the class of distributions considered. The error in the second regime is known as the asymptotic bias and describes the unavoidable error due to adversarial contamination. Therefore,

39th Conference on Neural Information Processing Systems (NeurIPS 2025).

such bias characterizes the optimal error for each class of distributions. Table 1 shows the order of the asymptotic bias corresponding to different classes of distributions and mean estimators.

Table 1: Order of the asymptotic bias of the optimal estimation error for different classes of distributions and mean estimators. Expressions in blue denote that the order is optimal in the corresponding class, while empty cells indicate unknown results. We denote by $\alpha$ the fraction of contaminated samples.

| | MoM | Trimmed Mean | M-estimator |
|---|---|---|---|
| Finite Variance: $\mathcal{P}_2$ | $\sqrt{\alpha}$ [this paper] | $\sqrt{\alpha}$ [24] | $\sqrt{\alpha}$ [25] |
| Infinite Variance: $\mathcal{P}_{1+r}$ | $\alpha^{\frac{r}{1+r}}$ [this paper] | | $\alpha^{\frac{r}{1+r}}$ [25] |
| Sub-Gaussian: $\mathcal{P}_{\mathrm{SG}}$ | $\alpha^{2/3}$ [this paper] | $\alpha\sqrt{\log(1/\alpha)}$ [24] | |
| Symmetric: $\mathcal{P}_{\mathrm{sym}}^{\varepsilon_0,c}$ | $\alpha$ [this paper] | | |
| Gaussian: $\mathcal{P}_{\mathrm{G}}$ | $\alpha$ e.g., [26, 27] | $\alpha\sqrt{\log(1/\alpha)}$ [24] | |

In the class of distributions with finite variance, the optimal order of the asymptotic bias is $\sqrt{\alpha}$, and both the trimmed mean and Catoni's M-estimator are known to be optimal in this class [24, 25]. In the class of distributions with infinite variance and finite absolute $(1 + r)$-th moment, the optimal order of the asymptotic bias becomes $\alpha^{\frac{r}{1+r}}$, and Catoni's M-estimator is known to be optimal in this class [25]. In the class of sub-Gaussian distributions, the optimal asymptotic bias improves to $\alpha\sqrt{\log(1/\alpha)}$, and the trimmed mean is known to be optimal in this class [24]. The asymptotic bias can be further improved to $\alpha$ in the class of Gaussian distributions [28], but the trimmed mean is known to be sub-optimal in this class [24].

Previous work has theoretically shown the suitability of the MoM estimator in certain contamination scenarios. In particular, the results in [15] provide upper bounds for MoM's error for finite-variance distributions, with rates matching the optimal order in the i.i.d. scenario. In addition, MoM has been shown to be (minimax) optimal for Gaussian distributions since it generalizes the sample median (see e.g., [26], [27, Cor. 1.15]). However, the (minimax) optimality of MoM remains unknown beyond the Gaussian case, as mentioned in [25]. Specifically, previous work on MoM considered a more benign contamination model and only studied the regime with a reduced number of samples, i.e., not the asymptotic bias. Moreover, the limitations of MoM under adversarial contamination remain unknown, as no lower bounds on its error have been established.

This paper provides upper and lower error bounds for the MoM estimator for multiple classes of distributions under adversarial contamination (see Table 1). In particular, the results reveal that MoM is (minimax) optimal for heavy-tailed and symmetric distributions, but sub-optimal for light-tailed distributions. Specifically, the main contributions in the paper are as follows:

- We prove that MoM is optimal under adversarial contamination in the class of distributions with finite variance, as well as in the class of distributions with infinite variance and finite absolute $(1 + r)$-th moment.

- We obtain upper bounds for the error of MoM in the classes of sub-exponential and sub-Gaussian distributions, which improve upon those established for the finite variance case.

- We obtain lower bounds for MoM that match the order of the presented upper bounds. In particular, we prove that MoM cannot fully exploit light tails and is sub-optimal for sub-exponential distributions.

- We prove that MoM is optimal under adversarial contamination in a class of symmetric distributions that includes Gaussians and heavy-tailed distributions such as Student's $t$.

The rest of this paper is organized as follows. Section 2 describes the problem of mean estimation under adversarial contamination. In Section 3, we present the optimality results for distributions with finite variance, and infinite variance with finite absolute $(1 + r)$-th moment. In Section 4, we present error bounds for MoM for light-tailed distributions, where we obtain improved orders compared to the finite variance case. In Section 5, we prove that MoM attains even better orders in a class of symmetric distributions. All proofs are deferred to Appendices A and B. Finally, we illustrate the results in the paper with numerical experiments in Appendix C.

**Notations.** For a real number $x$, $\lceil x \rceil$ denotes the smallest integer greater than or equal to $x$, and $\lfloor x \rfloor$ denotes the greatest integer less than or equal to $x$. For $s \geq 1$, $\mathcal{P}_s$ denotes the set of all probability distributions with finite absolute $s$-th moment. Given a set of real numbers $x_1, x_2, \ldots, x_n$, we denote by $x_{(1)}, x_{(2)}, \ldots, x_{(n)}$ their non-decreasing rearrangement. The notation $x \lesssim y$ describes cases where there exists a constant $c$ such that $x \leq cy$. In addition, the notation $x \asymp y$ describes cases where $x \lesssim y$ and $y \lesssim x$.

## 2 Preliminaries

This section begins by formulating the problem of one-dimensional mean estimation under adversarial contamination. We then discuss (minimax) optimal mean estimators and conclude by recalling the definition of the MoM estimator.

### 2.1 Contamination model

Let $X_1^*, X_2^*, \ldots, X_n^* \in \mathbb{R}$ be i.i.d. samples drawn from a distribution p with finite mean $\mu_\mathrm{p}$ and variance $\sigma_\mathrm{p}^2$. For some contamination fraction $\alpha$, the adversary changes at most $\alpha n$ of the samples, and the resulting contaminated samples are denoted as $X_1, X_2, \ldots, X_n$. This situation is referred to as adversarial contamination, and can be mathematically modeled as follows.

**Definition 2.1** (Adversarial contamination). We say that $X_1, X_2, \ldots, X_n$ are $\alpha$-contaminated samples if there exist i.i.d. samples $X_1^*, X_2^*, \ldots, X_n^*$ and indexes $1 \leq i_1 < i_2 < \cdots < i_r \leq n$ with $r \leq (1 - \alpha)n$, so that

$$\{X_{i_1}, X_{i_2}, \ldots, X_{i_r}\} \subseteq \{X_1^*, X_2^*, \ldots, X_n^*\}. \tag{1}$$

Adversarial contamination generalizes the additive contamination model (also called $\mathcal{O} \cup \mathcal{I}$ model). In the additive case, the contaminated samples $X_1, X_2, \ldots, X_n$ contain $(1 - \alpha)n$ inliers drawn independently from p together with $\alpha n$ contaminated samples, for which no specific assumption is made. Adversarial contamination describes more general and harmful types of attacks than the additive model. Conceptually, the distinction between the two contamination models can be understood as follows: under adversarial contamination, the adversary can selectively discard at most $\alpha n$ of the samples after inspecting their values, whereas in the additive model, the adversary discards samples randomly. Mathematically, under adversarial contamination, the samples $\{X_{i_1}, X_{i_2}, \ldots, X_{i_r}\}$ in (1) just need to be contained in $\{X_1^*, X_2^*, \ldots, X_n^*\}$, whereas in the additive contamination model they also have to be independent.

### 2.2 Minimax optimal mean estimators

The goal of mean estimation under adversarial contamination is to approximate the mean $\mu_\mathrm{p}$ using an estimator $\widehat{\mu} = \widehat{\mu}(X_1, X_2, \ldots, X_n)$ evaluated at $\alpha$-contaminated samples. Given a confidence parameter $\delta > 0$, we seek to guarantee that, with probability at least $1 - \delta$

$$|\widehat{\mu} - \mu_\mathrm{p}| \leq \sigma_\mathrm{p} \varepsilon(n, \delta, \alpha)$$

where $\varepsilon(n, \delta, \alpha)$ is as small as possible while guaranteeing the bound holds uniformly over all distributions p $\in \mathcal{P}$, where $\mathcal{P}$ is a class of distributions.

Previous work on mean estimation under adversarial contamination has shown that in the class $\mathcal{P}_2$ (distributions with finite variance), with probability $1 - \delta$, the estimation error is at best proportional to

$$\sigma_\mathrm{p} \left( \sqrt{\frac{\log(2/\delta)}{n}} + \sqrt{\alpha} \right). \tag{2}$$

Specifically, the results in [24] show that (2) is the (minimax) optimal error in $\mathcal{P}_2$. Namely, for any estimator $\widehat{\mu}$ of the mean, there exists a distribution p $\in \mathcal{P}_2$ and an adversarial attack such that with probability $1 - \delta$, the error $|\widehat{\mu} - \mu_\mathrm{p}|$ is lower bounded by (2), up to multiplicative constants. In addition, the expression (2) also provides an upper bound for some estimator $\widehat{\mu}$ for every p $\in \mathcal{P}_2$ and any adversarial attack.

The order of the optimal error in equation (2) exhibits two regimes depending on the sample size [24]: (1) for a reduced enough number of samples (when $n \leq \log(2/\delta)/\alpha$), the order of the bound takes the form $\sqrt{\log(2/\delta)/n}$, which matches the optimal order in the i.i.d. scenario; (2) as the number of samples increases (when $n \geq \log(2/\delta)/\alpha$), the optimal order plateaus at $\sqrt{\alpha}$, which is known as the asymptotic bias because it does not decrease with the number of samples. The error in the second regime describes the inevitable error caused by adversarial contamination.

For distributions with infinite variance, there is a similar optimality result [25]. Specifically, if $\mathcal{P}_{1+r}$ is the class of distributions with finite absolute $(1 + r)$-th moment, the optimal estimation error in $\mathcal{P}_{1+r}$ is proportional to

$$v_r^{\frac{1}{1+r}} \left( \left( \frac{\log(2/\delta)}{n} \right)^{\frac{r}{1+r}} + \alpha^{\frac{r}{1+r}} \right) \tag{3}$$

where $v_r = \mathbb{E}_{X \sim \mathrm{p}} |X - \mu_{\mathrm{p}}|^{1+r}$ denotes the absolute $(1 + r)$-th central moment of $\mathrm{p} \in \mathcal{P}_{1+r}$. As in the finite-variance case, the first term of the optimal error (3) matches the optimal error in the i.i.d. scenario for $\mathcal{P}_{1+r}$, whereas the second term $\alpha^{\frac{r}{1+r}}$ is the asymptotic bias.

The optimal asymptotic bias has different orders depending on the class of distributions considered (see Table 1 for a summary). In the class $\mathcal{P}_2$, both the trimmed mean and Catoni's M-estimator are known to be optimal [24, 25]. In addition, in the class $\mathcal{P}_{1+r}$, Catoni's M-estimator is known to be optimal [25]. If one considers a more restrictive class of distributions $\mathcal{P} \subset \mathcal{P}_2$, the order of the optimal asymptotic bias can be improved. For example, in the class of sub-Gaussian distributions $\mathcal{P}_{\mathrm{SG}}$, the optimal asymptotic bias is given by $\alpha\sqrt{\log(1/\alpha)}$, and the trimmed mean is optimal in this class [24]. In addition, in the class of Gaussian distributions $\mathcal{P}_{\mathrm{G}}$, the optimal asymptotic bias is $\alpha$ [28]. However, the trimmed mean is known to be sub-optimal in $\mathcal{P}_{\mathrm{G}}$ [24].

## 2.3 Median-of-Means

Let $X_1, X_2, \ldots, X_n$ be random samples and $I_1, I_2, \ldots, I_k$ be $k$ random disjoint blocks of the indices $\{1, 2 \ldots, n\}$ with equal size $m = \lfloor n/k \rfloor$.[1] Then, the MoM estimator is defined as the median of the means corresponding to different blocks, that is

$$\widehat{\mu}_{\mathrm{MoM}} = \widehat{\mu}_{(\lceil k/2 \rceil)}$$

where

$$\widehat{\mu}_i = \frac{1}{m} \sum_{j \in I_i} X_j, \quad i = 1, 2, \ldots, k$$

and $\widehat{\mu}_{(\lceil k/2 \rceil)}$ denotes the $\lceil k/2 \rceil$-th order statistic.

In the following sections, we present concentration inequalities for MoM under adversarial contamination for different classes of distributions.

# 3 Optimality of the MoM estimator for heavy-tailed distributions

In this section, we establish the optimality of the MoM estimator under adversarial contamination for general classes of distributions. We first define the quantile function which describes the behavior of averages of i.i.d. inliers.

**Definition 3.1.** Let $X_1^*, X_2^*, \ldots, X_m^*$ be $m$ i.i.d. random variables with distribution $\mathrm{p}$ and finite mean $\mu_{\mathrm{p}}$. We denote by $Q_m$ the quantile function of the random variable

$$B_m = \frac{X_1^* + X_2^* + \cdots + X_m^*}{m} - \mu_{\mathrm{p}}.$$

That is, $Q_m(q) = \inf\{x \in \mathbb{R} : q \leq \mathbb{P}[B_m \leq x]\}$ for $q \in [0, 1]$.

---

[1]As is commonly done for the analysis of the MoM estimator, by taking the floor function we ensure that all the blocks have the same size even when $k$ does not divide $n$. In practice, some blocks can have larger sizes to use all the samples, but this does not affect the theoretical results presented.

The bounds presented in the paper are consequences of the next theorem that shows how MoM's error is described by the behavior of averages of i.i.d. inliers around their median. In contrast, the error in other robust estimators is given by the behavior of the tails of the distribution [24].

**Theorem 3.1.** *Let $\widehat{\mu}_{\mathrm{MoM}}$ be the MoM estimator with $k$ blocks of size $m = \lfloor n/k \rfloor$ evaluated at $n$ $\alpha$-contaminated samples. If the number of blocks satisfies $2\alpha n < k \leq n$, then for all $\delta > 2\exp(-2k(1/2 - \alpha m)^2)$*

$$|\widehat{\mu}_{\mathrm{MoM}} - \mu_{\mathrm{p}}| \leq \max\left\{Q_m\left(\frac{1}{2} + \sqrt{\frac{\log(2/\delta)}{2k}} + \alpha m\right), -Q_m\left(\frac{1}{2} - \sqrt{\frac{\log(2/\delta)}{2k}} - \alpha m\right)\right\}$$

(4)

*holds with probability at least $1 - \delta$.*

*Sketch of proof.* The full proof is given in Appendix A.1.

Without loss of generality, we assume $\mu_{\mathrm{p}} = 0$. Let $\{\widehat{\mu}_i^*\}_{i=1}^k$ be the sample means of the different blocks before the attack of the adversary, and let $\{\widehat{\mu}_i\}_{i=1}^k$ be the sample means of the different blocks after the adversary's attack. By the definition of adversarial contamination, the adversary can modify at most $\alpha n$ samples, which affects at most $\alpha n$ of the block means. Therefore, the sets $\{\widehat{\mu}_i^*\}_{i=1}^k$ and $\{\widehat{\mu}_i\}_{i=1}^k$ differ in at most $\alpha n$ elements.

Since MoM is the median of the block means, i.e., $\widehat{\mu}_{\mathrm{MoM}} = \widehat{\mu}_{(\lceil k/2 \rceil)}$, we have $\widehat{\mu}_{(\lceil k/2 \rceil - \alpha n)}^* \leq \widehat{\mu}_{\mathrm{MoM}} \leq \widehat{\mu}_{(\lceil k/2 \rceil + \alpha n)}^*$. Then, the result in (4) follows since the empirical quantiles $\widehat{\mu}_{(\lceil k/2 \rceil - \alpha n)}^*$ and $\widehat{\mu}_{(\lceil k/2 \rceil + \alpha n)}^*$ are close to the actual quantiles $Q_m(1/2 + \alpha m)$ and $Q_m(1/2 - \alpha m)$. $\qquad\square$

The result above presents a bound for the error of MoM under adversarial contamination that is valid with wide generality. In the next subsections, we derive from Theorem 3.1 the optimal bounds for the different classes of distributions, adjusting the block size. This is achieved by bounding the quantile function $Q_m$ around $1/2$, uniformly over the class of distributions. A general approach to obtain a reduced bound for the right-hand side of (4) is to increase the blocks' size $m$. For instance, if p has finite variance, we have the bound

$$Q_m(1/2 + \varepsilon) \lesssim \frac{1}{\sqrt{m(1/2 - \varepsilon)}}$$

(5)

as a consequence of Chebyshev's inequality, for any $\varepsilon \in [0, 1/2)$. Better bounds for (4) can be obtained if the distribution p enjoys certain symmetry. For instance, if p is a Gaussian distribution, for any $\varepsilon \in [0, 1/2)$,

$$Q_m(1/2 + \varepsilon) \lesssim \frac{\varepsilon}{\sqrt{m}}$$

(6)

and we can obtain better bounds than (5) by decreasing $\varepsilon$ towards zero.

## 3.1 Finite variance

The next result establishes the optimality of MoM under adversarial contamination in the class $\mathcal{P}_2$ formed by distributions with finite variance.

**Theorem 3.2.** *Let $\widehat{\mu}_{\mathrm{MoM}}$ be the MoM estimator with a number of blocks*

$$k = \max\left\{\left\lceil \frac{\log(2/\delta)}{(1/2 - 1/\gamma)^2}\right\rceil, \lceil \gamma\alpha n\rceil\right\}$$

(7)

*for any $\gamma \in (2, 2.5]$ and $\delta > 2\exp(-(1/2 - 1/\gamma)^2 n)$. Then, there exists a positive constant $C(\gamma) \asymp (1/2 - 1/\gamma)^{-3/2}$ such that for any p $\in \mathcal{P}_2$ and $\alpha \leq 0.4$*

$$|\widehat{\mu}_{\mathrm{MoM}} - \mu_{\mathrm{p}}| \leq C(\gamma)\sigma_{\mathrm{p}}\left(\sqrt{\frac{\log(2/\delta)}{n}} + \sqrt{\alpha}\right)$$

(8)

*holds with probability at least $1 - \delta$.*

*Sketch of proof.* The full proof is given in Appendix A.2.

The key step is to bound the quantile function $Q_m$ appearing in Theorem 3.1. Since p has finite variance, Chebysev's inequality implies that, for any $\varepsilon \in [0, 1/2)$

$$Q_m(1/2 + \varepsilon) \leq \frac{\sigma_{\mathrm{p}}}{\sqrt{m(1/2 - \varepsilon)}}.$$

The same bound also holds for $-Q_m(1/2-\varepsilon)$. Thus, both quantiles of interest scale as $\mathcal{O}(\sigma_{\mathrm{p}}/\sqrt{m})$.

For the choice of $k$ in the theorem, namely

$$k \asymp \log(2/\delta) + \alpha n$$

we get

$$m \asymp n/(\log(2/\delta) + \alpha n).$$

Substituting this into the Chebyshev bound gives

$$\frac{\sigma_{\mathrm{p}}}{\sqrt{m(1/2 - \varepsilon)}} \lesssim \sigma_{\mathrm{p}} \left( \sqrt{\frac{\log(2/\delta)}{n}} + \sqrt{\alpha} \right).$$

Finally, applying Theorem 3.1, which relates the performance of the MoM estimator to bounds on $Q_m$, we conclude that

$$|\widehat{\mu}_{\mathrm{MoM}} - \mu_{\mathrm{p}}| \lesssim \sigma_{\mathrm{p}} \left( \sqrt{\frac{\log(2/\delta)}{n}} + \sqrt{\alpha} \right) \tag{9}$$

with probability at least $1 - \delta$. $\qquad\square$

The result above shows that MoM is (minimax) optimal in the class $\mathcal{P}_2$. Moreover, Theorem 3.2 generalizes existing robustness results for MoM under heavy tails. Specifically, without contamination (i.e., $\alpha = 0$) the bound (8) recovers the known sub-Gaussianity result for MoM in the i.i.d. scenario [10]. Theorem 3.2 also generalizes existing results for MoM in scenarios with contaminated samples. Specifically, the results in [15] provide upper bounds for MoM's error with rates that match the optimal order in the i.i.d. scenario. However, such results are limited to scenarios with additive contamination and only characterize the first optimality regime. In particular, existing bounds apply when the number of samples satisfies $n \lesssim \log(2/\delta)/\alpha$, and therefore do not characterize the asymptotic bias (see e.g., [15, Prop. 2]).

Theorem 3.2 shows that the contamination tolerance of MoM is comparable to other estimators. In fact, requiring a maximum tolerance for $\alpha$ is standard for mean estimators [24, 25]. For instance, existing results establish a contamination tolerance of $\alpha \leq 0.13$ for the trimmed mean, and of $\alpha \leq 0.36$ for Catoni's M-estimator [24, 25].

The best known leading constants in the error bounds for the i.i.d. scenario are significantly smaller than the one in Theorem 3.2 (e.g., the bound for Lee-Valiant estimator in the i.i.d. scenario has a leading constant of $\sqrt{2}$ [14]). As in other works for adversarial contamination [24], the focus of this paper is on establishing optimal rates rather than minimizing constants.

## 3.2 Infinite variance

The next result shows that MoM is also optimal in the class $\mathcal{P}_{1+r}$ of distributions whose absolute $(1 + r)$-th moment $v_r$ is finite for $r \in (0, 1)$.

**Theorem 3.3.** *Let $\widehat{\mu}_{\mathrm{MoM}}$ be the MoM estimator with a number of blocks*

$$k = \max\left\{ \left\lceil \frac{\log(2/\delta)}{(1/2 - 1/\gamma)^2} \right\rceil, \lceil \gamma \alpha n \rceil \right\} \tag{10}$$

*for any $\gamma \in (2, 2.5]$ and $\delta > 2\exp(-(1/2 - 1/\gamma)^2 n)$. Then, there exists a constant $C(\gamma) \asymp (1/2 - 1/\gamma)^{-\frac{2r+1}{1+r}}$ such that for any $\mathrm{p} \in \mathcal{P}_{1+r}$ and $\alpha \leq 0.4$*

$$|\widehat{\mu}_{\mathrm{MoM}} - \mu_{\mathrm{p}}| \leq C(\gamma) v_r^{\frac{1}{1+r}} \left( \left( \frac{\log(2/\delta)}{n} \right)^{\frac{r}{1+r}} + \alpha^{\frac{r}{1+r}} \right) \tag{11}$$

*holds with probability at least $1 - \delta$.*

*Sketch of proof.* The full proof is given in Appendix A.3.

The argument parallels Theorem 3.2. The key step is to bound the quantile function $Q_m$ in Theorem 3.1. By combining Markov's inequality with the Bahr-Esseen inequality [29, Thm. 2], for any $\varepsilon \in [0, 1/2)$ we get

$$Q_m(1/2 + \varepsilon) \lesssim \left( \frac{v_r}{m^r(1/2 - \varepsilon)} \right)^{\frac{1}{1+r}}$$

and the same bound also holds for $-Q_m(1/2 - \varepsilon)$. Hence, both quantiles of interest scale as $\mathcal{O}((v_r/m^r)^{\frac{1}{1+r}})$.

With the choice of $k$ in the theorem, namely:

$$k \asymp \log(2/\delta) + \alpha n$$

we have

$$m \asymp n/(\log(2/\delta) + \alpha n).$$

Plugging this into the quantile bound yields

$$\left( \frac{v_r}{m^r(1/2 - \varepsilon)} \right)^{\frac{1}{1+r}} \lesssim v_r^{\frac{1}{1+r}} \left( \left( \frac{\log(2/\delta)}{n} \right)^{\frac{r}{1+r}} + \alpha^{\frac{r}{1+r}} \right).$$

Finally, invoking Theorem 3.1 we conclude that

$$|\widehat{\mu}_{\mathrm{MoM}} - \mu_{\mathrm{p}}| \lesssim v_r^{\frac{1}{1+r}} \left( \left( \frac{\log(2/\delta)}{n} \right)^{\frac{r}{1+r}} + \alpha^{\frac{r}{1+r}} \right)$$

with probability at least $1 - \delta$. $\square$

Theorem 3.3 presents the first analysis of MoM for distributions with infinite variance in the presence of contamination. Moreover, our bound shows that the error of MoM attains the optimal order in the class $\mathcal{P}_{1+r}$ shown in [25]. Theorem 3.3 also generalizes existing robustness results for MoM under heavy tails. Specifically, without contamination (i.e., $\alpha = 0$) the bound (11) recovers the optimality result of MoM for infinite variance in the i.i.d. scenario [8].

In terms of contamination tolerance, Theorem 3.2 shows that MoM presents a notable improvement compared to Catoni's M-estimator, the other optimal estimator in the class $\mathcal{P}_{1+r}$. In particular, the tolerance to contamination of Catoni's M-estimator decreases with $r$. For example, for values of $r = 0.5$ and $r = 0.1$, Catoni's M-estimator can handle contamination levels up to $\alpha = 0.26$ and $\alpha = 0.16$, respectively [25]. Theorem 3.3 shows that the tolerance to contamination of MoM does not decrease with $r$.

### 3.3 Matching lower bounds for general distributions

In this section, we have established the optimality of MoM in $\mathcal{P}_2$ and $\mathcal{P}_{1+r}$ for the same choice of the number of blocks $k$. The next theorem shows that, for this choice of $k$, the $\sqrt{\alpha}$ order of the asymptotic bias cannot be improved for any distribution $\mathrm{p} \in \mathcal{P}_3$ with a finite third absolute moment.[2]

**Theorem 3.4.** *Let $\widehat{\mu}_{\mathrm{MoM}}$ be the MoM estimator with $k$ given as in (7) and (10). There exist positive constants $C, \alpha_{\max}$, and an adversarial attack such that for any $\mathrm{p} \in \mathcal{P}_3$ and $\alpha < \alpha_{\max}$*

$$|\widehat{\mu}_{\mathrm{MoM}} - \mu_{\mathrm{p}}| \geq C\sigma_{\mathrm{p}}\sqrt{\alpha} \tag{12}$$

*holds with constant probability.*

---

[2]We thank an anonymous reviewer for pointing out that the finiteness of the third moment is not essential for a result similar to that in Theorem 3.4. In particular, an asymptotic result can be derived using the central limit theorem without requiring a finite third moment.

*Sketch of proof.* The full proof is given in Appendix B.1.

Let $\widehat{\mu}_1^*, \widehat{\mu}_2^*, \ldots, \widehat{\mu}_k^*$ denote the empirical means of the $k$ blocks before contamination and $B$ be the set of the $\lceil k/2 \rceil$ blocks with lowest sample means. With constant probability, the number of blocks in $B$ that contain at least one contaminated sample exceeds $\alpha n / C_1$, for some universal constant $C_1 > 0$ (Lemma B.1). Therefore, with that probability, MoM is shifted to

$$\widehat{\mu}_{\mathrm{MoM}} = \widehat{\mu}^*_{(\lceil k/2 \rceil + \lceil \alpha n / C_1 \rceil)}$$

if the adversary includes samples with arbitrarily large values. In addition, for sufficiently large $n$, the empirical quantile $\widehat{\mu}^*_{(\lceil k/2 \rceil + \lceil \alpha n / C_1 \rceil)}$ is arbitrarily close to the actual quantile $Q_m(1/2 + \alpha m / C_1)$ for $m = \lfloor n/k \rfloor$. Then, the result is obtained since $m \asymp 1/\alpha$ and the Berry-Essen theorem implies $Q_m(1/2 + \alpha m / C_1) \gtrsim \sigma_{\mathrm{p}} / \sqrt{m} \gtrsim \sigma_{\mathrm{p}} \sqrt{\alpha}$. $\qquad\square$

The result above shows one limitation of MoM under adversarial contamination: choosing the number of blocks $k$ to achieve (minimax) optimal error rates over a broad class of heavy-tailed distributions does not lead to improved rates for specific, well-behaved distributions. Specifically, for every distribution with finite variance (e.g., Gaussian), the estimation error has the same order as in the worst case. Therefore, the number of blocks must be adjusted differently depending on the class of distributions considered. A similar limitation affects Catoni's M-estimator, which requires selecting an appropriate function $\psi$ (a parameter that determines the estimator) depending on the distribution considered [25]. In contrast, other robust estimators such as the trimmed mean [24] do not need to adjust parameters depending on the distribution.

In the following sections, we present error bounds for MoM that yield better orders for the asymptotic bias than $\sqrt{\alpha}$ considering specific subclasses $\mathcal{P} \subsetneq \mathcal{P}_2$. In line with the above discussion, this improvement necessitates a different choice for the number of blocks $k$.

## 4 Sub-optimality of the MoM estimator for light-tailed distributions

In this section, we show that the asymptotic bias can be improved for light-tailed distributions. In particular, we prove that the asymptotic bias of MoM is upper bounded by $\alpha^{2/3}$ for sub-exponential distributions. In addition, we provide matching lower bounds showing that MoM is sub-optimal for light-tailed distributions. This result complements known limitations of MoM in the i.i.d. scenario, such as its large asymptotic variance [9].

### 4.1 Sub-exponential distributions

The following result presents a concentration inequality for MoM which holds in the class of sub-exponential distributions denoted as $\mathcal{P}_{\mathrm{SE}}$. Distributions $\mathrm{p} \in \mathcal{P}_{\mathrm{SE}}$ are characterized by the condition that, for all $s \geq 2$, $(\mathbb{E}_{X \sim \mathrm{p}} |X - \mu_{\mathrm{p}}|^s)^{1/s} \leq c \sigma_{\mathrm{p}} s$ [30, Prop. 2.7.1].

**Theorem 4.1.** *Let $\widehat{\mu}_{\mathrm{MoM}}$ be the MoM estimator with a number of blocks $k = \lceil \xi \alpha^{2/3} n \rceil$ for any $\xi > 0$ and $\delta > 2 \exp(-\xi n^{1/3} / 18)$. There exist positive constants $C(\xi)$ and $\alpha_{\max}(\xi)$ such that for any $\mathrm{p} \in \mathcal{P}_{\mathrm{SE}}$ and $\alpha < \alpha_{\max}(\xi)$*

$$|\widehat{\mu}_{\mathrm{MoM}} - \mu_{\mathrm{p}}| \leq C(\xi) \sigma_{\mathrm{p}} \left( \sqrt{\frac{\log(2/\delta)}{n}} + \alpha^{2/3} \right) \tag{13}$$

*holds with probability at least $1 - \delta$.*

*Proof.* See Appendix A.4. $\qquad\square$

The result above shows that the asymptotic bias of MoM can attain better orders than $\sqrt{\alpha}$ (as established in Theorem 3.2) when restricting to the class of sub-exponential distributions. However, the confidence parameter in Theorem 4.1 must satisfy $\delta > 2 \exp(-\mathcal{O}(n^{1/3}))$, whereas Theorem 3.2 allows for the broader range of values $\delta > 2 \exp(-\mathcal{O}(n))$. Such limitations are common in estimators that do not depend on $\delta$ (multiple-$\delta$ estimators) [31].

The asymptotic bias shown in Theorem 4.1 is far from the optimal order in the class $\mathcal{P}_{\mathrm{SE}}$, which cannot be higher than $\alpha \log(1/\alpha)$ because the trimmed mean estimator achieves such an order (see

remark in page 10 of [24]). Notably, in the next theorem, we show that for any MoM, the order of $\alpha^{2/3}$ cannot be improved in the class $\mathcal{P}_{\mathrm{SE}}$.

**Theorem 4.2.** *There exist positive constants $C, \alpha_{\max}$, a probability distribution $\mathrm{p} \in \mathcal{P}_{\mathrm{SE}}$ and an adversarial attack such that for any $\alpha < \alpha_{\max}$ and for a MoM estimator $\widehat{\mu}_{\mathrm{MoM}}$ with any number of blocks $k \in \{1, 2, \ldots, n\}$,*

$$|\widehat{\mu}_{\mathrm{MoM}} - \mu_{\mathrm{p}}| \geq C\sigma_{\mathrm{p}}\alpha^{2/3}$$

*holds with constant probability*

*Proof.* See Appendix B.2. $\qquad\square$

The result above shows that MoM is (minimax) sub-optimal in the class $\mathcal{P}_{\mathrm{SE}}$. Specifically, Theorem 4.2 shows that for some sub-exponential distributions, the error of MoM for any choice of $k$ is lower bounded by an expression with order $\alpha^{2/3}$ that matches the upper bound in Theorem 4.1 and is sub-optimal in $\mathcal{P}_{\mathrm{SE}}$.

The suboptimality of MoM can be understood through the role of asymmetry in the error achieved using the median to estimate the mean. While MoM's performance improves under light-tailed distributions (Theorem 4.1), MoM does not attain the optimal rate for all such distributions, since some of them are quite asymmetric. In particular, there exist light-tailed distributions where the median differs substantially from the mean. Since MoM estimates the mean by taking the median (a biased estimator of the mean) of $k$ sample means (each an unbiased estimator), the bias becomes more pronounced in asymmetric distributions.

## 4.2 Sub-Gaussian distributions

The class of sub-Gaussian distributions $\mathcal{P}_{\mathrm{SG}}$ contains distributions whose tails decay at least as fast as those of a Gaussian. Distributions $\mathrm{p} \in \mathcal{P}_{\mathrm{SG}}$ are characterized by the condition that, for all $s \geq 2$, $(\mathbb{E}_{X \sim \mathrm{p}}|X - \mu_{\mathrm{p}}|^s)^{1/s} \leq c\sigma_{\mathrm{p}}\sqrt{s}$ [30, Prop. 2.5.2].

The results in Theorem 4.1 also provide an upper bound for sub-Gaussian distributions since $\mathcal{P}_{\mathrm{SG}} \subseteq \mathcal{P}_{\mathrm{SE}}$. As we experimentally show in Appendix C, the order $\alpha^{2/3}$ cannot be improved across all sub-Gaussian distributions. Nevertheless, in the following we show that MoM can attain better orders than $\alpha^{2/3}$ for some classes of sub-Gaussian distributions with certain symmetry. However, these orders do not match the optimal asymptotic bias in $\mathcal{P}_{\mathrm{SG}}$, which is of the order $\alpha\sqrt{\log(1/\alpha)}$, as shown in [24].

**Definition 4.1.** We define $\mathcal{P}_{\mathrm{SG}}^s$ for any integer $s \geq 3$ to be the set of absolutely continuous distributions $\mathrm{p} \in \mathcal{P}_{\mathrm{SG}}$ such that for all $j \leq s$

$$\mathbb{E}_{X \sim \mathrm{p}}\left[\left(\frac{X - \mu_{\mathrm{p}}}{\sigma_{\mathrm{p}}}\right)^j\right] = \mathbb{E}_{Z \sim N(0,1)}\left[Z^j\right]$$

where $N(0, 1)$ is a standard Gaussian distribution.

For any distribution in the class $\mathcal{P}_{\mathrm{SG}}^s$ the first $s$ central odd moments are zero, indicating that the distribution has certain symmetry around its mean. The next result shows that MoM can attain better orders for the asymptotic bias in the subclasses $\mathcal{P}_{\mathrm{SG}}^s$.

**Theorem 4.3.** *Let $\widehat{\mu}_{\mathrm{MoM}}$ be the MoM estimator with a number of blocks $k = \lceil \xi\alpha^{\frac{2}{s+1}}n \rceil$ for any $\xi > 0$ and $\delta > 2\exp\left(-\xi n^{\frac{s-1}{s+1}}/18\right)$. There exist positive constants $C(\xi)$ and $\alpha_{\max}(\xi)$ such that for any $\mathrm{p} \in \mathcal{P}_{\mathrm{SG}}^s$ and $\alpha < \alpha_{\max}(\xi)$*

$$|\widehat{\mu}_{\mathrm{MoM}} - \mu_{\mathrm{p}}| \leq C(\xi)\sigma_{\mathrm{p}}\left(\sqrt{\frac{\log(2/\delta)}{n}} + \alpha^{\frac{s}{s+1}}\right) \qquad (14)$$

*holds with probability at least $1 - \delta$.*

*Proof.* See Appendix A.5. $\qquad\square$

Increasing $s$ results in an increased symmetry that leads to an asymptotic bias in (14) approaching $\alpha$. In the following section, we show that MoM can attain an asymptotic bias of order $\alpha$ for some symmetric distributions that include the Gaussians and even distributions with heavy tails.

# 5 Optimality of the MoM estimator for symmetric distributions

In this section, we show that the asymptotic bias can be further improved to $\alpha$ for symmetric distributions with quantile function that increases at most linearly around $1/2$. We denote by $\mathcal{P}_{\mathrm{sym}}$ the set of symmetric distributions around its mean, i.e., distributions p such that $X - \mu_{\mathrm{p}}$ and $-(X - \mu_{\mathrm{p}})$ have the same distribution, whenever $X \sim \mathrm{p}$.

**Definition 5.1.** We define the class of distributions $\mathcal{P}_{\mathrm{sym}}^{\varepsilon_0,c}$, for $\varepsilon_0 < 1/3$ and $c > 5$, to be the set of $\mathrm{p} \in \mathcal{P}_{\mathrm{sym}} \cap \mathcal{P}_2$ such that for all $\varepsilon \in [0, \varepsilon_0]$ and $m \in \mathbb{N}$

$$Q_m(1/2 + \varepsilon) \leq c \frac{\sigma_{\mathrm{p}}}{\sqrt{m}} \varepsilon \tag{15}$$

where $Q_m$ is the quantile function from Definition 3.1 corresponding to the distribution p.

The class $\mathcal{P}_{\mathrm{sym}}^{\varepsilon_0,c}$ contains Gaussian distributions and heavy-tailed distributions like Student's $t$ distributions. This is easy to check since both Gaussian and Student's $t$ distributions are symmetric, satisfy (15) for $m = 1$, and are infinite divisible distributions [32]. In the literature, similar classes to $\mathcal{P}_{\mathrm{sym}}^{\varepsilon_0,c}$ have been analyzed in the context of quantile regression and mean estimation with outliers. For example, $\mathcal{P}_{\mathrm{sym}}^{\varepsilon_0,c}$ is similar to the distributions with $1/2$-quantile of type 2 from [33, Def. 2.1] and to the class of symmetric distributions defined in [34].

The technical condition (15) ensures that the quantile function is not excessively sharp near the median (it increases at most linearly), so that it is easier to distinguish the median from nearby quantiles. More precisely, the constant $\varepsilon_0$ controls the size of the neighborhood in which the quantile function increases at most linearly, whereas the constant $c$ controls the slope of the linear approximation.

The following result shows that MoM is optimal in the class $\mathcal{P}_{\mathrm{sym}}^{\varepsilon_0,c}$ under adversarial contamination.

**Theorem 5.1.** *Let $\widehat{\mu}_{\mathrm{MoM}}$ be the MoM estimator with $k = \lceil \beta n \rceil$ blocks for any $\beta \leq 1$ and $\delta > 2\exp(-\beta\varepsilon_0^2 n/4)$. Then, there exists a positive constant $C = C(c, \beta)$ such that for any $\mathrm{p} \in \mathcal{P}_{\mathrm{sym}}^{\varepsilon_0,c}$ and $\alpha < \beta\varepsilon_0/2$*

$$|\widehat{\mu}_{\mathrm{MoM}} - \mu_{\mathrm{p}}| \leq C\sigma_{\mathrm{p}} \left( \sqrt{\frac{\log(2/\delta)}{n}} + \alpha \right) \tag{16}$$

*holds with probability at least $1 - \delta$.*

*Proof.* The result follows directly from Theorem 3.1 and Definition 5.1. $\qquad\square$

The result above establishes the (minimax) optimality of MoM under adversarial contamination in the class $\mathcal{P}_{\mathrm{sym}}^{\varepsilon_0,c}$, for any pair $\varepsilon_0, c$. Theorem 5.1 generalizes the well-known guarantee for the sample median (case $\beta = 1$) for the class of Gaussian distributions (see e.g., [26], [27, Cor. 1.15]).

# 6 Conclusion

This paper provides upper and lower bounds on the error of the MoM estimator for multiple classes of distributions under adversarial contamination. Specifically, we prove that MoM is (minimax) optimal in the class of distributions with finite variance, and in the class of distributions with finite absolute $(1 + r)$-th moment (infinite variance). In addition, we show that the MoM estimator is particularly well-suited for symmetric distributions. These results reinforce the widely recognized strengths of MoM, such as its optimality in the i.i.d. scenario. On the other hand, the paper also reveals that MoM has certain limitations under adversarial contamination. In particular, we show that MoM cannot fully leverage light-tails, and we characterize its sub-optimality for sub-exponential distributions. These results complement known limitations of MoM in the i.i.d. scenario, such as its large asymptotic variance. Overall, the theoretical results presented in the paper provide a comprehensive characterization of the capabilities of the MoM estimator under adversarial contamination.

## Acknowledgments

The authors would like to thank Prof. Gábor Lugosi for his comments and suggestions during the development of this work. Funding in direct support of this work has been provided by project PID2022-137063NB-I00 funded by MCIN/AEI/10.13039/501100011033 and the European Union "NextGenerationEU"/PRTR, BCAM Severo Ochoa accreditation CEX2021-001142-S/MICIN/AEI/10.13039/501100011033 funded by the Ministry of Science and Innovation (Spain), and program BERC-2022-2025 funded by the Basque Government. Xabier de Juan acknowledges a predoctoral grant from the Basque Government.

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

# A  Proofs for Upper Bounds

The following result is an adaptation of Proposition 2a from [35] that is used to prove Theorem 3.1. For clarity, we assume that p is an absolutely continuous distribution. This assumption allows the cumulative distribution function to have an inverse, which corresponds to the quantile function. This assumption is not restrictive and the proofs can be adapted to include any distribution with finite variance (see [35, Prop. 2a] for similar arguments in the context of conformal prediction). An alternative solution for the statistician is to add a small amount of independent Gaussian noise to the samples, ensuring that the distribution has a density without compromising statistical performance.

**Lemma A.1.** *Let* p *be a distribution with quantile function $Q$, and let $X_1, X_2, \ldots, X_n$ be $n$ i.i.d. samples drawn from* p. *For $r \in \{1, 2, \ldots, n\}$ and $\varepsilon = r/n$, the following holds:*

- *If $\varepsilon + \sqrt{\log(1/\delta)/2n} \in (0,1)$, then*

$$X_{(r)} \leq Q\left(\varepsilon + \sqrt{\frac{\log(1/\delta)}{2n}}\right) \tag{17}$$

  *with probability at least $1 - \delta$.*

- *If $\varepsilon - \sqrt{\log(1/\delta)/2n} \in (0,1)$, then*

$$X_{(r)} \geq Q\left(\varepsilon - \sqrt{\frac{\log(1/\delta)}{2n}}\right) \tag{18}$$

  *with probability at least $1 - \delta$.*

*Proof.* Let $F$ be the CDF of p and let $t \in (0,1)$ be a real number to be chosen later. Since the $X_1, X_2, \ldots, X_n$ are independent

$$\mathbb{P}[X_{(r)} > Q(t)] = \mathbb{P}\left[\sum_{i=1}^{n} \mathbb{I}\{X_i \leq Q(t)\} \leq r - 1\right] = \mathbb{P}[\mathrm{Bin}(n, F(Q(t)) \leq r - 1]$$

where, for any $m \in \mathbb{N}$, $q \in [0,1]$, $\mathrm{Bin}(m, q)$ denotes a random variable drawn from a binomial distribution of $m$ trials with a probability of success $q$. Taking $t = \varepsilon + \sqrt{\log(1/\delta)/(2n)}$, we apply Hoeffding's inequality (as $\varepsilon - t < 0$) and obtain

$$\mathbb{P}[\mathrm{Bin}(n, t) \leq r - 1] \leq \mathbb{P}[\mathrm{Bin}(n, t) \leq \varepsilon n]$$
$$\leq \mathbb{P}[\mathrm{Bin}(n, t)/n - t \leq \varepsilon - t] \leq e^{-2(t-\varepsilon)^2 n} = \delta$$

that leads to (17) since we have shown $\mathbb{P}[X_{(r)} > Q(t)] \leq \delta$.

Now we prove (18) in a similar way to (17). Analogously to the previous case, we have

$$\mathbb{P}[X_{(r)} < Q(t)] = \mathbb{P}\left[\sum_{i=1}^{n} \mathbb{I}\{X_i \leq Q(t)\} \geq r\right] = \mathbb{P}[\mathrm{Bin}(n, F(Q(t))) \geq r]$$

so that taking $t = \varepsilon - \sqrt{\log(1/\delta)/(2n)}$, and applying Hoeffding's inequality (since $\varepsilon - t > 0$) we get

$$\mathbb{P}[\mathrm{Bin}(n, t) \geq r] = \mathbb{P}[\mathrm{Bin}(n, t) \geq \varepsilon n]$$
$$\leq \mathbb{P}[\mathrm{Bin}(n, t)/n - t \geq \varepsilon - t] \leq e^{-2(\varepsilon-t)^2 n} = \delta$$

that leads to (18). $\qquad\square$

## A.1 Proof of Theorem 3.1

*Proof of Theorem 3.1.* Without loss of generality, we assume $\mu_p = 0$. Let $I_1, I_2, \ldots, I_k$ be $k$ random disjoint blocks of the indices $\{1, 2 \ldots, n\}$ of equal size $m = \lfloor n/k \rfloor$. Let

$$\widehat{\mu}_i^* = \frac{1}{m} \sum_{j \in I_i} X_j^*, \quad i = 1, 2, \ldots, k \tag{19}$$

be the sample means corresponding to different blocks before the adversary contaminates the sample and let

$$\widehat{\mu}_i = \frac{1}{m} \sum_{j \in I_i} X_j, \quad i = 1, 2, \ldots, k \tag{20}$$

be the sample means corresponding to different blocks after the adversary contaminates the sample.

The attack of the adversary can modify at most the values of $\alpha n$ blocks, so $\widehat{\mu}_{\mathrm{MoM}} = \widehat{\mu}_{(\lceil k/2 \rceil)}$ is between $\widehat{\mu}_{(\lceil k/2 \rceil - \alpha n)}^*$ and $\widehat{\mu}_{(\lceil k/2 \rceil + \alpha n)}^*$, which are well defined since $k > 2\alpha n$.

Since $\sqrt{\log(2/\delta)/(2k)} + \alpha m < 1/2$ by hypothesis, applying Lemma A.1,

$$\widehat{\mu}_{(\lceil k/2 \rceil + \alpha n)}^* \le Q_m \left( \frac{1}{2} + \sqrt{\frac{\log(2/\delta)}{2k}} + \alpha m \right) =: u$$

holds with probability at least $1 - \delta/2$, and also

$$l := Q_m \left( \frac{1}{2} - \sqrt{\frac{\log(2/\delta)}{2k}} - \alpha m \right) \le \widehat{\mu}_{(\lceil k/2 \rceil + \alpha n)}^*$$

holds with probability at least $1 - \delta/2$. Moreover, by the union bound

$$\min\{l, -u\} \le l \le \widehat{\mu}_{\mathrm{MoM}} \le u \le \max\{u, -l\}$$

holds with probability at least $1 - \delta$. Since $\min\{l, -u\} = -\max\{u, -l\}$

$$|\widehat{\mu}_{\mathrm{MoM}}| \le \max \left\{ Q_m \left( \frac{1}{2} + \sqrt{\frac{\log(2/\delta)}{2k}} + \alpha m \right), -Q_m \left( \frac{1}{2} - \sqrt{\frac{\log(2/\delta)}{2k}} - \alpha m \right) \right\}$$

holds with probability at least $1 - \delta$. $\qquad\qquad\square$

## A.2 Proof of Theorem 3.2

*Proof of Theorem 3.2.* Without loss of generality, we assume $\mu_p = 0$. By Theorem 3.1

$$|\widehat{\mu}_{\mathrm{MoM}}| \le \max \left\{ Q_m \left( \frac{1}{2} + \sqrt{\frac{\log(2/\delta)}{2k}} + \alpha m \right), -Q_m \left( \frac{1}{2} - \sqrt{\frac{\log(2/\delta)}{2k}} - \alpha m \right) \right\}$$

holds with probability $1 - \delta$ whenever

$$\sqrt{\frac{\log(2/\delta)}{2k}} + \alpha m < 1/2. \tag{21}$$

Some basic computations verify that the choice of $k$ in the hypothesis satisfies $2\alpha n < k \le n$ and that (21) holds for the range of $\delta$ specified in the hypothesis.

For every $\varepsilon \in [0, 1/2)$, if $Q = Q_m(1/2 + \varepsilon)$, by Markov's inequality we have

$$1/2 - \varepsilon = \mathbb{P}[B_m \ge Q] = \mathbb{P}[B_m^2 \ge Q^2] \le \frac{\sigma_p^2}{mQ^2} \tag{22}$$

and in particular,

$$Q_m(1/2 + \varepsilon) \le \frac{\sigma_p}{\sqrt{m}} \frac{1}{\sqrt{1/2 - \varepsilon}}.$$

Moreover, we also have

$$-Q_m(1/2 - \varepsilon) \le \frac{\sigma_{\mathrm{p}}}{\sqrt{m}} \frac{1}{\sqrt{1/2 - \varepsilon}}$$

since $-Q_m(1/2 - \varepsilon) = Q_{-m}(1/2 + \varepsilon)$, where $Q_{-m}$ is the quantile function of $-B_m$.

Let $c = (1/2 - 1/\gamma)^2$ be a positive constant. If $\delta \in (2e^{-cn}, 2e^{-c\gamma\alpha n})$, the number of blocks is given by $k = \lceil \log(2/\delta)/c \rceil$. In particular, $1/\sqrt{m} \le \sqrt{2k/n} \le 2\sqrt{\log(2/\delta)/(cn)}$, since $\lfloor x \rfloor \ge x/2$ and $\lceil x \rceil \le 2x$ for any $x \ge 1$. Therefore, if $\varepsilon = \sqrt{\log(2/\delta)/(2k)} + \alpha m$,

$$\frac{\sigma_{\mathrm{p}}}{\sqrt{m}} \frac{1}{\sqrt{1/2 - \varepsilon}} \le c_1' \sigma_{\mathrm{p}} \sqrt{\frac{\log(2/\delta)}{n}}$$

where

$$c_1' \le 2\sqrt{\frac{1/c}{1/2 - (\sqrt{c/2} + 1/\gamma)}}.$$

If $\delta \ge 2e^{-\gamma\alpha n}$, the number of blocks equals $k = \lceil \gamma\alpha n \rceil$, and in particular, $1/\sqrt{m} \le 2\sqrt{\gamma\alpha}$. Hence, if $\varepsilon = \sqrt{\log(2/\delta)/(2k)} + \alpha m$,

$$\frac{\sigma_{\mathrm{p}}}{\sqrt{m}} \frac{1}{\sqrt{1/2 - \varepsilon}} \le c_1'' \sigma_{\mathrm{p}} \sqrt{\alpha}$$

where

$$c_1'' \le 2\sqrt{\frac{\gamma}{1/2 - (\sqrt{c/2} + 1/\gamma)}}.$$

Therefore, we have shown that for all $\delta > e^{-cn}$ and $\gamma \le 2.5 = 1/0.4 \le 1/\alpha$, if $k = \max\{\lceil \gamma\alpha n \rceil, \lceil \log(2/\delta)/c \rceil\}$

$$|\widehat{\mu}_{\mathrm{MoM}}| \le C\sigma_{\mathrm{p}} \left( \sqrt{\frac{\log(2/\delta)}{n}} + \sqrt{\alpha} \right)$$

holds with probability at least $1 - \delta$, where

$$C = 2\sqrt{2 + \sqrt{2}} \left( \frac{1}{(1/2 - 1/\gamma)^{3/2}} + \frac{\sqrt{\gamma}}{(1/2 - 1/\gamma)^{1/2}} \right). \tag{23}$$

$\square$

## A.3 Proof of Theorem 3.3

*Proof.* Without loss of generality, we assume $\mu_{\mathrm{p}} = 0$. By Theorem 3.1

$$|\widehat{\mu}_{\mathrm{MoM}}| \le \max\left\{ Q_m\left( \frac{1}{2} + \sqrt{\frac{\log(2/\delta)}{2k}} + \alpha m \right), -Q_m\left( \frac{1}{2} - \sqrt{\frac{\log(2/\delta)}{2k}} - \alpha m \right) \right\}$$

holds with probability $1 - \delta$, whenever

$$\sqrt{\frac{\log(2/\delta)}{2k}} + \alpha m < 1/2. \tag{24}$$

Some basic computations verify that the choice of $k$ in the hypothesis satisfies $2\alpha n < k \le n$ and that (24) holds for the range of $\delta$ specified in the hypothesis.

For all $\varepsilon \in [0, 1/2)$, by Markov's inequality (following a similar reasoning as in (22)),

$$Q_m(1/2 + \varepsilon) \le \left( \frac{\mathbb{E}|B_m|^{1+r}}{1/2 - \varepsilon} \right)^{\frac{1}{1+r}}.$$

Moreover, by the Bahr-Esseen inequality [29, Thm. 2], moments of i.i.d. averages satisfy $\mathbb{E}|B_m|^{1+r} \leq 2m^{-r}v_r$. Therefore, we have

$$Q_m(1/2 + \varepsilon) \leq 2^{\frac{1}{1+r}} \frac{v_r^{\frac{1}{1+r}}}{m^{\frac{r}{1+r}}} \frac{1}{(1/2 - \varepsilon)^{\frac{1}{1+r}}}.$$

Furthermore, we get

$$-Q_m(1/2 - \varepsilon) \leq 2^{\frac{1}{1+r}} \frac{v_r^{\frac{1}{1+r}}}{m^{\frac{r}{1+r}}} \frac{1}{(1/2 - \varepsilon)^{\frac{1}{1+r}}}$$

since $-Q_m(1/2 - \varepsilon) = Q_{-m}(1/2 + \varepsilon)$, where $Q_{-m}$ denotes the quantile function of $-B_m$.

The rest of the proof proceeds analogously to the one for Theorem 3.2 in Appendix A.2. Let $c = (1/2 - 1/\gamma)^2$ be a positive constant. If $\delta \in (2e^{-cn}, 2e^{-c\gamma\alpha n})$, the number of blocks is given by $k = \lceil \log(2/\delta)/c \rceil$, and if $\varepsilon = \sqrt{\log(2/\delta)/(2k)} + \alpha m$,

$$2^{\frac{1}{1+r}} \frac{v_r^{\frac{1}{1+r}}}{m^{\frac{r}{1+r}}} \frac{1}{(1/2 - \varepsilon)^{\frac{1}{1+r}}} \leq c_1' v_r^{\frac{1}{1+r}} \left( \frac{\log(2/\delta)}{n} \right)^{\frac{r}{1+r}} \tag{25}$$

where $c_1'$ depends only on $\gamma$ and $r$.

If $\delta \geq 2e^{-\gamma\alpha n}$, the number of blocks equals $k = \lceil \gamma\alpha n \rceil$, and if $\varepsilon = \sqrt{\log(2/\delta)/(2k)} + \alpha m$,

$$2^{\frac{1}{1+r}} \frac{v_r^{\frac{1}{1+r}}}{m^{\frac{r}{1+r}}} \frac{1}{(1/2 - \varepsilon)^{\frac{1}{1+r}}} \leq c_1'' v_r^{\frac{1}{1+r}} \alpha^{\frac{r}{1+r}} \tag{26}$$

where $c_1''$ depends only on $\gamma$ and $r$. Combining both (25) and (26) we get the desired result. $\qquad\square$

## A.4 Proof of Theorem 4.1

Minsker introduced in [9] a technique to study the error of MoM in terms of the rates of convergence in normal approximations. We extend this approach to the scenario of adversarial contamination. In particular, the definition bellow describes the difference between averages of i.i.d. inliers and the standard Gaussian distribution.

**Definition A.1.** For every $m \in \mathbb{N}$ we define $g(m) \geq 0$ as

$$g(m) = \sup_{t \in \mathbb{R}} \left| F_m \left( \frac{\sigma_p t}{\sqrt{m}} \right) - \Phi(t) \right|$$

where $F_m$ is the cumulative density function from Definition 3.1, and $\Phi$ is the cumulative density function of a standard Gaussian distribution $N(0, 1)$.

In the following lemma, we combine Theorem 3.1 with the ideas in Section 2 from [9] to obtain bounds for the error of MoM under adversarial contamination in terms of $g(m)$.

**Lemma A.2.** *Let $\widehat{\mu}_{\mathrm{MoM}}$ be the MoM estimator with $k$ blocks of size $m = \lfloor n/k \rfloor$ evaluated at $n$ $\alpha$-contaminated samples. If the number of blocks satisfies $2\alpha n < k \leq n$, there exists a universal constant $C$ such that for all $\delta > 2\exp(-2k(1/3 - \alpha m - g(m))^2)$*

$$|\widehat{\mu}_{\mathrm{MoM}} - \mu_p| \leq C\sigma_p \left( \sqrt{\frac{\log(2/\delta)}{n}} + \alpha\sqrt{m} + \frac{g(m)}{\sqrt{m}} \right)$$

*holds with probability at least $1 - \delta$.*

*Proof.* We first prove that for all $\varepsilon \in [0, 1/2 - g(m))$

$$-Q_m(1/2 - \varepsilon) \leq \frac{\sigma_p}{\sqrt{m}} \Phi^{-1}(1/2 + \varepsilon + g(m)), \tag{27}$$

$$Q_m(1/2 + \varepsilon) \leq \frac{\sigma_p}{\sqrt{m}} \Phi^{-1}(1/2 + \varepsilon + g(m)). \tag{28}$$

We only prove (28), since (27) is obtained in the symmetric way. Note that $\Phi(t) - F_m\left(\sigma_{\mathrm{p}} t / \sqrt{m}\right) \leq g(m)$ for all $t \in \mathbb{R}$. In particular, if $t = \Phi^{-1}(1/2 + \varepsilon + g(m))$

$$1/2 + \varepsilon = \Phi(t) - g(m) \leq F_m\left(\frac{\sigma_{\mathrm{p}} t}{\sqrt{m}}\right).$$

Therefore, we get

$$Q_m(1/2 + \varepsilon) \leq \frac{\sigma_{\mathrm{p}}}{\sqrt{m}} t = \frac{\sigma_{\mathrm{p}}}{\sqrt{m}} \Phi^{-1}(1/2 + \varepsilon + g(m)).$$

Since $\sqrt{\log(2/\delta)/2k} + \alpha m + g(m) < 1/3$ by hypothesis, by Theorem 3.1

$$|\widehat{\mu}_{\mathrm{MoM}} - \mu_{\mathrm{p}}| \leq \frac{\sigma_{\mathrm{p}}}{\sqrt{m}} \Phi^{-1}\left(\frac{1}{2} + \sqrt{\frac{\log(2/\delta)}{2k}} + \alpha m + g(m)\right)$$

holds with probability at least $1 - \delta$, after applying both (27) and (28). Moreover, it is well known that $\Phi^{-1}(1/2 + z) \leq 3z$ for all $z < 1/3$ [9, Lemma 4]. Therefore, there exists a universal constant $C > 0$ such that

$$|\widehat{\mu}_{\mathrm{MoM}} - \mu_{\mathrm{p}}| \leq C\sigma_{\mathrm{p}}\left(\sqrt{\frac{\log(2/\delta)}{n}} + \alpha\sqrt{m} + \frac{g(m)}{\sqrt{m}}\right)$$

holds with probability at least $1 - \delta$, as desired. $\qquad\square$

*Proof of Theorem 4.1.* Let $\alpha_{\max} = \min\{\xi^{-3/2}, \xi^3/8, (6(\xi^{-1} + 2\tilde{C}_3\sqrt{\xi}))^{-3}\}$ be the tolerance to contamination, where $\tilde{C}_3$ is a constant specified later. It is easy to check that $2\alpha n < k \leq n$ is satisfied for all $\xi > 0$, since $\alpha < \alpha_{\max}$.

By the Berry-Esseen theorem [36, Sec. 1.2], the value of $g(m)$ in Definition A.1 satisfies $g(m) \leq \tilde{C}_1 \rho_{\mathrm{p}}/(\sigma_{\mathrm{p}}^3 \sqrt{m})$ for some constant $\tilde{C}_1 > 0$ and where $\rho_{\mathrm{p}} = \mathbb{E}_{X \sim \mathrm{p}}|X - \mu_{\mathrm{p}}|^3$. Moreover, since $\mathrm{p}$ is sub-exponential $\rho_{\mathrm{p}} \leq \tilde{C}_2 \sigma_{\mathrm{p}}^3$ for some universal constant $\tilde{C}_2 > 0$. Therefore, there is a constant $\tilde{C}_3 > 0$ such that $g(m) \leq \tilde{C}_3/\sqrt{m}$. Hence, if $\delta > 2\exp(-2k(1/3 - \alpha m - \tilde{C}_3/\sqrt{m})^2)$, by Lemma A.2

$$|\widehat{\mu}_{\mathrm{MoM}} - \mu_{\mathrm{p}}| \leq C_1\sigma_{\mathrm{p}}\left(\sqrt{\frac{\log(2/\delta)}{n}} + \alpha\sqrt{m} + \frac{1}{m}\right)$$

holds with probability at least $1 - \delta$, for a positive constant $C_1$.

By hypothesis $k = \lceil \xi \alpha^{2/3} n \rceil$, and since $m = \lfloor n/k \rfloor$,

$$|\widehat{\mu}_{\mathrm{MoM}} - \mu_{\mathrm{p}}| \leq C_2\sigma_{\mathrm{p}}\left(\sqrt{\frac{\log(2/\delta)}{n}} + \alpha^{2/3}\right)$$

holds with probability at least $1 - \delta$, for a constant $C_2 = C_2(\xi)$ whenever $\delta > 2\exp(-2k(1/3 - \alpha m - \tilde{C}_3/\sqrt{m})^2)$. Without loss of generality, we assume $\alpha \geq 1/n$. Therefore the result also holds for all $\delta > 2\exp(-\xi n^{1/3}/18)$ since $\alpha \in [1/n, (6(\xi^{-1} + 2\tilde{C}_3\sqrt{\xi}))^{-3})$ implies $2\exp(-\xi n^{1/3}/18) > 2\exp(-2k(1/3 - \alpha m - \tilde{C}_3/\sqrt{m})^2)$. $\qquad\square$

## A.5  Proof of Theorem 4.3

*Proof of Theorem 4.3.* Since the proof is almost identical to that of Theorem 4.1 in Appendix A.4, we omit the computations and provide only a sketch of the proof. Instead of using Berry-Essen theorem, we now use Theorem 1.1. from [37] to upper bound $g(m)$. Therefore, since $\mathrm{p} \in \mathcal{P}_{\mathrm{SG}}^{(s)}$, by [37, Thm 1.1.]

$$g(m) \leq \tilde{C}_1 \frac{\mathbb{E}_{X \sim \mathrm{p}}|X - \mu_{\mathrm{p}}|^{s+1}}{\sigma_{\mathrm{p}}^{s+1} m^{\frac{s-1}{2}}}$$

for some constant $\tilde{C}_1 > 0$. In particular, as p is sub-Gaussian, there exists a constant $\tilde{C}_2 > 0$ such that $\mathbb{E}_{X \sim p}|X - \mu_p|^{s+1} \leq \tilde{C}_2 \sigma_p^{s+1}$. Thus, there is a constant $\tilde{C}_3 > 0$ such that $g(m) \leq \tilde{C}_3/m^{\frac{s-1}{2}}$. Therefore, for $\delta > 2\exp(-2k(1/3 - \alpha m - \tilde{C}_3/m^{s-1})^2)$, by Lemma A.2

$$|\widehat{\mu}_{\text{MoM}} - \mu_p| \leq C_1 \sigma_p \left( \sqrt{\frac{\log(2/\delta)}{n}} + \alpha\sqrt{m} + \frac{1}{m^{s/2}} \right)$$

holds with probability at least $1 - \delta$, for a constant $C_1 > 0$.

By hypothesis, $k = \lceil \xi \alpha^{\frac{2}{s+1}} n \rceil$, and since $m = \lfloor n/k \rfloor$,

$$|\widehat{\mu}_{\text{MoM}} - \mu_p| \leq C_2 \sigma_p \left( \sqrt{\frac{\log(2/\delta)}{n}} + \alpha^{\frac{s}{s+1}} \right)$$

holds with probability at least $1 - \delta$, for a constant $C_2 > 0$. Proceeding as in the proof of Theorem 4.1 in Appendix A.4, we get the desired lower bound for $\delta$.

$\square$

# B  Proofs for Lower Bounds

The following two auxiliary lemmas are used to prove the lower bounds in theorems 3.4 and 4.2.

**Lemma B.1.** *Let $C_1 > 3$ and $Z_\alpha$ be the number of blocks with at least one contaminated sample among the $\lceil k/2 \rceil$ blocks with lowest sample means $\widehat{\mu}_1^*, \widehat{\mu}_2^*, \ldots, \widehat{\mu}_k^*$ before contamination. Then there exists an adversarial attack and $C_2 > 0$ such that for all $\alpha \geq 1/n$*

$$\mathbb{P}[Z_\alpha > \alpha n/C_1] > C_2. \tag{29}$$

*Proof.* Note that $Z_\alpha = \sum_{i=1}^{\lceil k/2 \rceil} Y_i$, where the random variable $Y_i \in \{0, 1\}$ is defined as $Y_i = 1$ if the $i$-th block, ranked by the smallest sample means before contamination, contains at least one contaminated sample, and $Y_i = 0$ otherwise.

We consider a contamination in which the adversary randomly chooses $\alpha n$ samples, and makes them arbitrarily large. Hence, all the $Y_i$ are identically distributed.

Note that for all $i \neq j$, $Y_i$ and $Y_j$ are negatively correlated, i.e., $\text{Cov}(Y_i, Y_j) \leq 0$. To see this, since all $Y_1, Y_2, \ldots, Y_k$ have the same mean, we get $\text{Cov}(Y_i, Y_j) = \mathbb{E}[Y_i Y_j] - (\mathbb{E}Y_i)^2$. Let $\mathcal{C}_i$ denote the event such that $Y_i = \mathbb{I}\{\mathcal{C}_i\}$. Since $\mathbb{P}[\mathcal{C}_i|\mathcal{C}_j] \leq \mathbb{P}[\mathcal{C}_i]$ and the $Y_i$ are identically distributed

$$\mathbb{E}[Y_i Y_j] = \mathbb{P}[\mathcal{C}_i \cap \mathcal{C}_j] = \mathbb{P}[\mathcal{C}_j]\mathbb{P}[\mathcal{C}_i|\mathcal{C}_j] \leq \mathbb{P}[\mathcal{C}_j]\mathbb{P}[\mathcal{C}_i] = (\mathbb{E}Y_i)^2$$

whence we get $\text{Cov}(Y_i, Y_j) \leq 0$.

Therefore, we can apply to $Z_\alpha$ the Chernoff bound for negatively correlated random variables [38]. That is, for all $\theta \in (0, 1)$

$$\mathbb{P}[Z_\alpha > (1 - \theta)\mathbb{E}Z_\alpha] \geq 1 - \exp\left( -\frac{\theta^2 \mathbb{E}Z_\alpha}{2} \right). \tag{30}$$

We can lower bound the r.h.s. of (30) if we find a lower bound for $\mathbb{E}Z_\alpha$. We claim $\mathbb{E}Z_\alpha > 1 - e^{-1/2}$, and prove it in the following lines. Let $f(\alpha)$ be the probability that all the samples in a block are not contaminated. Then it is easy to see that for all $i \in \{1, 2, \ldots, k\}$, $\mathbb{E}Y_i = 1 - f(\alpha)$ where

$$f(\alpha) = \frac{\binom{n - \alpha n}{m}}{\binom{n}{m}}, \quad m = \left\lfloor \frac{n}{k} \right\rfloor.$$

Moreover, $f(\alpha) \leq e^{-\alpha m}$ since

$$f(\alpha) = \prod_{j=0}^{m-1} \frac{n - \alpha n - j}{n - j} \leq \left( \frac{n - \alpha n}{n} \right)^m = (1 - \alpha)^m \leq e^{-\alpha m}. \tag{31}$$

Finally,

$$\mathbb{E}Z_\alpha \geq \left\lceil \frac{\lceil n/m \rceil}{2} \right\rceil (1 - e^{-\alpha m}) \geq \frac{1}{2\alpha m}(1 - e^{-\alpha m}) > 1 - e^{-1/2} \tag{32}$$

where in the first inequality we have applied (31), in the second one $n \geq 1/\alpha$ and in the last one $m < 1/(2\alpha)$. Therefore, applying $\mathbb{E}Z_\alpha > 1 - e^{-1/2}$ in (30) we get

$$1 - \exp\left(-\frac{\theta^2 \mathbb{E}Z_\alpha}{2}\right) > 1 - \exp\left(-\frac{\theta^2(1 - e^{-1/2})}{2}\right) = C_2.$$

We can upper bound the l.h.s. of (30) as follows,

$$\mathbb{P}[Z_\alpha > \alpha n/C_1] \geq \mathbb{P}[Z_\alpha > (1 - \theta)\mathbb{E}Z_\alpha]$$

where $1/(C_1(1 - e^{-1/2})) \leq (1 - \theta)$. To see this, note that the following chain of inequalities hold

$$\frac{\alpha n}{C_1} \leq \alpha n(1 - \theta)(1 - e^{-1/2})$$

$$\leq \alpha n(1 - \theta)\frac{1}{2\alpha m}(1 - e^{-\alpha m})$$

$$\leq (1 - \theta)\mathbb{E}Z_\alpha$$

where in the second inequality we have used $m < 1/(2\alpha)$ and in the last one $\mathbb{E}Z_\alpha \geq n(1 - e^{-\alpha m})/(2m)$, as in (32). $\qquad \square$

**Lemma B.2.** *Let $C_1 > 3$ be a constant, let $\widehat{\mu}_{\mathrm{MoM}}$ be the MoM estimator with blocks of size $m$, and let $F_m$ be the c.d.f. of $B_m$ in Definition 3.1. There exists an adversarial attack such that for any distribution $p$ with finite mean, and any $t > 0$ such that $F_m(t) \leq 1/2 + \alpha m/C_1$,*

$$|\widehat{\mu}_{\mathrm{MoM}} - \mu_p| \geq t \tag{33}$$

*holds with constant probability.*

*Proof.* Without loss of generality we assume $\mu_p = 0$. Let $C_1 > 3$ and $C_2$ be the constant from Lemma B.1. By the law of total probability

$$\mathbb{P}[|\widehat{\mu}_{\mathrm{MoM}}| > t] \geq \mathbb{P}[|\widehat{\mu}_{\mathrm{MoM}}| > t | Z_\alpha > \alpha n/C_1] \mathbb{P}[Z_\alpha > \alpha n/C_1].$$

We consider a contamination in which the adversary randomly chooses $\alpha n$ samples, and makes them arbitrarily large, as considered in Lemma B.1. Thus, by that lemma we have a lower bound for the second term $\mathbb{P}[Z_\alpha > \alpha n/C_1] > C_2$. Thus, it remains to lower bound the first term $\mathbb{P}[|\widehat{\mu}_{\mathrm{MoM}}| > t | Z_\alpha > \alpha n/C_1]$. First note that

$$\mathbb{P}[|\widehat{\mu}_{\mathrm{MoM}}| > t | Z_\alpha > \alpha n/C_1] \geq \mathbb{P}[|\widehat{\mu}^*_{(\lceil k/2 \rceil + \lceil \alpha n/C_1 \rceil)}| > t] \tag{34}$$

since there are at least $\lceil \alpha n/C_1 \rceil$ contaminated blocks not above the median before the attack.

We can further lower bound (34) with a binomial tail

$$\mathbb{P}[|\widehat{\mu}^*_{(\lceil k/2 \rceil + \lceil \alpha n/C_1 \rceil)}| > t] \geq \mathbb{P}[\widehat{\mu}^*_{(\lceil k/2 \rceil + \lceil \alpha n/C_1 \rceil)} > t]$$

$$= \mathbb{P}\left[\sum_{i=1}^{k} \mathbb{I}\{\widehat{\mu}^*_i \leq t\} < \left\lceil \frac{k}{2} \right\rceil + \left\lceil \frac{\alpha n}{C_1} \right\rceil\right]$$

$$= \mathbb{P}\left[\mathrm{Bin}(k, F_m(t)) < \left\lceil \frac{k}{2} \right\rceil + \left\lceil \frac{\alpha n}{C_1} \right\rceil\right]$$

$$\geq \mathbb{P}\left[\mathrm{Bin}(k, F_m(t)) \leq k\left(\frac{1}{2} + \frac{\alpha m}{C_1}\right)\right]. \tag{35}$$

Since the binomial tails are monotonically decreasing in the success parameter [35, Lemma 1], equation (35) can be further lower bounded as

$$\mathbb{P}\left[\mathrm{Bin}(k, F_m(t)) \leq k\left(\frac{1}{2} + \frac{\alpha m}{C_1}\right)\right] \geq \mathbb{P}\left[\mathrm{Bin}\left(k, \frac{1}{2} + \frac{\alpha m}{C_1}\right) \leq k\left(\frac{1}{2} + \frac{\alpha m}{C_1}\right)\right]$$

since $F_m(t) \leq 1/2 + \alpha m/C_1$ by hypothesis.

Write $q = 1/2 + \alpha m/C_1$ for simplicity. It is easy to check that $q < 1 - 1/k$, since $C_1$ can be taken as large as needed. Therefore, by Corollary 3 from [39] we have $\mathbb{P}[\mathrm{Bin}(k, q) \leq kq] \geq 1/4$, which translates to the desired lower bound $\mathbb{P}[|\widehat{\mu}_{\mathrm{MoM}}| > t | Z_\alpha > \alpha n/C_1] \geq 1/4$. $\qquad \square$

## B.1 Proof of Theorem 3.4

*Proof of Theorem 3.4.* Let $F_m$ be the c.d.f. of $B_m$ in Definition 3.1. Since $\mathrm{p} \in \mathcal{P}_3$, by the Berry-Essen theorem [36, Sec. 1.2], there exists $\tilde{C} > 0$ such that

$$\sup_{t \in \mathbb{R}} \left| F_m \left( \frac{\sigma_\mathrm{p} t}{\sqrt{m}} \right) - \Phi(t) \right| \leq \frac{\tilde{C}}{\sqrt{m}}$$

and in particular, for any $t \in \mathbb{R}$

$$F_m \left( \frac{\sigma_\mathrm{p} t}{\sqrt{m}} \right) \leq \Phi(t) + \frac{\tilde{C}}{\sqrt{m}}. \tag{36}$$

Let $C_1 > 3$, $\alpha_{\max} = (32 C_1 \tilde{C} \gamma^{3/2})^{-2}$ be the contamination tolerance, and $C = 1/(8 C_1 \sqrt{\gamma}) > 0$. In the following we prove $F_m(C \sigma_\mathrm{p} \sqrt{\alpha}) \leq 1/2 + \alpha m / C_1$ for all $\alpha \leq \alpha_{\max}$ so that by Lemma B.2 such bound is sufficient to prove the statement of Theorem 3.4.

Note that there exists a range for $\delta$ such that $k = \lceil \gamma \alpha n \rceil$, so that $4/(\gamma \alpha) \leq m \leq 1/(\gamma \alpha)$. For all $\alpha \leq \alpha_{\max}$

$$
\begin{aligned}
F_m \left( C \sigma_\mathrm{p} \sqrt{\alpha} \right) &\leq F_m \left( \frac{\sigma_\mathrm{p} C}{\sqrt{m \gamma}} \right) && (m \leq 1/(\gamma \alpha)) \\
&\leq \Phi \left( \frac{C}{\sqrt{\gamma}} \right) + \frac{\tilde{C}}{\sqrt{m}} && \text{(Apply (36))} \\
&\leq \frac{1}{2} + \frac{C}{\sqrt{\gamma}} + \frac{\tilde{C}}{\sqrt{m}} && (z \geq 0 \implies \Phi(z) \leq 1/2 + z) \\
&\leq \frac{1}{2} + \frac{C}{\sqrt{\gamma}} + 4 \tilde{C} \sqrt{\gamma \alpha_{\max}} && (m \geq 4/(\gamma \alpha) \text{ and } \alpha \leq \alpha_{\max}).
\end{aligned}
$$

Finally, since $C = 1/(8 C_1 \sqrt{\gamma})$, $\alpha_{\max} = (32 C_1 \tilde{C} \gamma^{3/2})^{-2}$ and $m \geq 4/(\gamma \alpha)$,

$$\frac{1}{2} + \frac{C}{\sqrt{\gamma}} + 4 \tilde{C} \sqrt{\gamma \alpha_{\max}} = \frac{1}{2} + \frac{1}{4 \gamma C_1} \leq \frac{1}{2} + \frac{\alpha m}{C_1}$$

so that $F_m(C \sigma_\mathrm{p} \sqrt{\alpha}) \leq 1/2 + \alpha m / C_1$, as desired.

$\square$

## B.2 Proof of Theorem 4.2

*Proof of Theorem 4.2.* The proof is divided in two parts depending on whether $m > 1/\alpha^{2/3}$ or $m \leq 1/\alpha^{2/3}$.

If $m > 1/\alpha^{2/3}$, we show that there exists an adversarial attack and $C > 0$ such that for any distribution $\mathrm{p} \in \mathcal{P}_{\mathrm{SE}}$

$$|\hat{\mu}_{\mathrm{MoM}} - \mu_\mathrm{p}| \geq C \sigma_\mathrm{p} \alpha \sqrt{m} \tag{37}$$

holds with constant probability. By Lemma B.2, establishing the lower bound in (37) reduces to proving $F_m(C \sigma_\mathrm{p} \alpha \sqrt{m}) \leq 1/2 + \alpha m / C_1$, where $C_1 > 3$. Note that since $m > 1/\alpha^{2/3}$, the lower bound in (37) can be further lower bounded to an order of $\alpha^{2/3}$.

Since $\mathrm{p} \in \mathcal{P}_{\mathrm{SE}}$, as a consequence of the Berry-Essen theorem [36, Sec. 1.2], there exists $\tilde{C} > 0$ such that for any $t \in \mathbb{R}$,

$$F_m \left( \frac{\sigma_\mathrm{p} t}{\sqrt{m}} \right) \leq \Phi(t) + \frac{\tilde{C}}{\sqrt{m}}. \tag{38}$$

Let $r \in (2/3, 1)$ be such that $m = 1/\alpha^r$ and let $\alpha_{\max} = (2 C_1 \tilde{C})^{-\frac{1}{3r/2 - 1}}$ be the contamination tolerance. In what follows, we will prove $F_m(C \sigma_\mathrm{p} \alpha \sqrt{m}) \leq 1/2 + \alpha m / C_1$ for all $\alpha \leq \alpha_{\max}$ with $C = 1/(2 C_1)$.

We have

$$F_m\left(C\sigma_{\mathrm{p}}\alpha\sqrt{m}\right) = F_m\left(\frac{\sigma_{\mathrm{p}}C\alpha m}{\sqrt{m}}\right)$$

$$\leq \Phi\left(C\alpha m\right) + \frac{\tilde{C}}{\sqrt{m}} \qquad \text{(Apply (38))}$$

$$\leq \frac{1}{2} + C\alpha m + \frac{\tilde{C}}{\sqrt{m}} \qquad (z \geq 0 \implies \Phi(z) \leq 1/2 + z).$$

Finally, since $\alpha \leq \alpha_{\max}$, $C = 1/(2C_1)$ and $r > 2/3$, it is straightforward to check

$$\frac{1}{2} + C\alpha m + \frac{\tilde{C}}{\sqrt{m}} = \frac{1}{2} + C\alpha^{1-r} + \tilde{C}\alpha^{r/2} \leq \frac{1}{2} + \frac{\alpha^{1-r}}{C_1} = \frac{1}{2} + \frac{\alpha m}{C_1}$$

so that $F_m(C\sigma_{\mathrm{p}}\sqrt{\alpha}) \leq 1/2 + \alpha m/C_1$, as desired.

If $m \leq 1/\alpha^{2/3}$ we show that if p is a negative exponential distribution then there exists an adversarial attack and $C > 0$ such that

$$|\widehat{\mu}_{\mathrm{MoM}} - \mu_{\mathrm{p}}| \geq C\sigma_{\mathrm{p}}\frac{1}{m} \tag{39}$$

holds with constant probability. By Lemma B.2, establishing the lower bound in (39) reduces to proving $F_m(C\sigma_{\mathrm{p}}/m) \leq 1/2 + \alpha m/C_1$, where $C_1 > 3$. Note that since $m \leq 1/\alpha^{2/3}$, the lower bound in (39) can be further lower bounded to an order of $\alpha^{2/3}$.

We consider distribution p corresponding to a negative exponential with parameter 1, i.e. $X \sim$ p if $-X \sim \mathrm{Exp}(1)$. It is straightforward to show that there exists $C > 0$ such that $C\sigma_{\mathrm{p}}/m \leq -\mu_{\mathrm{p}} - e^{-\frac{1}{3m}}$. The average of $m$ i.i.d. exponential distributions follows a Gamma distribution $\Gamma(m, 1/m)$ [40, p. 82], and the median of such Gamma is upper bounded by $e^{-\frac{1}{3m}}$ [41, Prop. 3.6]. Let $F$ denote the cumulative density function of a $\Gamma(m, 1/m)$. Since $-(B_m + \mu_{\mathrm{p}})$ follows a $\Gamma(m, 1/m)$,

$$\frac{1}{2} \leq F\left(e^{-\frac{1}{3m}}\right) = \mathbb{P}\left[-(B_m + \mu_{\mathrm{p}}) \leq e^{-\frac{1}{3m}}\right]$$

$$= \mathbb{P}\left[-\mu_{\mathrm{p}} - e^{-\frac{1}{3m}} \leq B_m\right] = 1 - F_m\left(-\mu_{\mathrm{p}} - e^{-\frac{1}{3m}}\right).$$

Therefore, $F_m\left(-\mu_{\mathrm{p}} - e^{-\frac{1}{3m}}\right) \leq 1/2$, and since $C\sigma_{\mathrm{p}}/m \leq -\mu_{\mathrm{p}} - e^{-\frac{1}{3m}}$, we get $F_m(C\sigma_{\mathrm{p}}/m) \leq 1/2 + \alpha m/C_1$, as desired.

$\square$

## C   Numerical Experiments

In this appendix, we illustrate the theoretical results in previous sections with numerical simulations. In particular, the experiments show that MoM performs particularly well for symmetric distributions but does not fully leverage light-tails in accordance with the theoretical results.

In all the results in this section, for a fixed estimator $\widehat{\mu}$ and distribution p, we repeated the following procedure $n_{\mathrm{rep}}$ times:

1. Draw $n$ i.i.d. samples from p.
2. Contaminate the i.i.d. sample, where $\alpha$ is the fraction of contaminated samples.
3. Compute the estimation error $|\widehat{\mu} - \mu_{\mathrm{p}}|$.

Finally, we plot the $1 - \delta$ quantile of the $n_{\mathrm{rep}}$ computed errors $|\widehat{\mu} - \mu_{\mathrm{p}}|$ as a function of $\alpha$. The empirical data is shown using dashed lines, while solid lines represent the graphs of the asymptotic bias when an upper bound is known. All plots are shown in log-log scale to clearly display the order of the asymptotic bias.

In table 2, we summarize the parameter of the various experimental results presented in the present section. In addition, we relate each figure to the corresponding theorem it illustrates.

The trimmed mean and Catoni's M-estimator have been implemented following the definitions and results in [24, 25]. For the MoM estimator, in figs. 1(a) and 1(b) we have used a number of blocks $k = \lceil 3\alpha n \rceil$ (corresponding to $\gamma = 3$ in Theorems 3.2 and 3.3), whereas in fig. 1(c) we set $k = \lceil n/5 \rceil$ (corresponding to $\beta = 1/5$ in Theorem 5.1).

In all the experimental results, we consider the following adversarial attack: given an uncontaminated sample $X_1^*, X_2^*, \ldots, X_n^*$ the adversary removes the $\alpha n$ largest values and replaces them with $\alpha n$ new samples, all set to $\min_i X_i^*$. Since the empirical error follows the same order as the upper bounds established in our theoretical results, no other attack can cause greater damage (in terms of order). However, it is true that some attacks may lead to larger absolute errors, although this would not change the error order.

The experimental results in the paper can be carried out in a regular desktop machine in few hours.

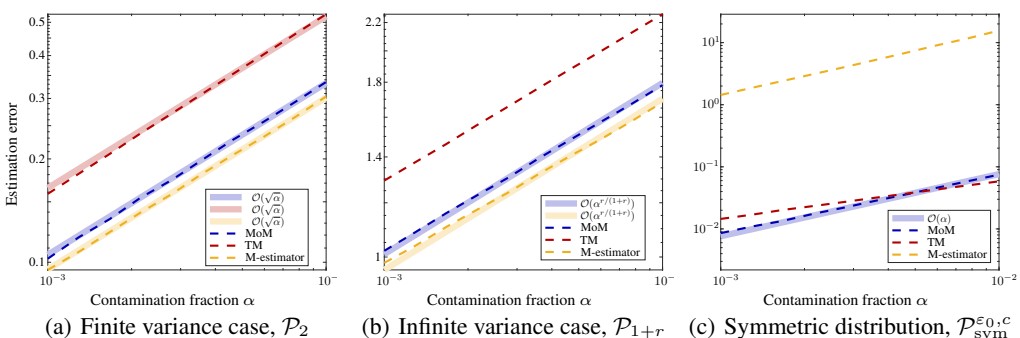

(a) Finite variance case, $\mathcal{P}_2$    (b) Infinite variance case, $\mathcal{P}_{1+r}$    (c) Symmetric distribution, $\mathcal{P}_{\text{sym}}^{\varepsilon_0,c}$

Figure 1: Empirical errors align with the theoretical bounds presented over multiple classes of distributions.

In fig. 1 we show that the empirical error of MoM aligns with the orders of the results from sections 3 to 5. We also provide the empirical errors of the trimmed mean and Catoni's M-estimator. The experiments in fig. 1 can be seen as an illustration of the first, second and fifth rows of Table 1.

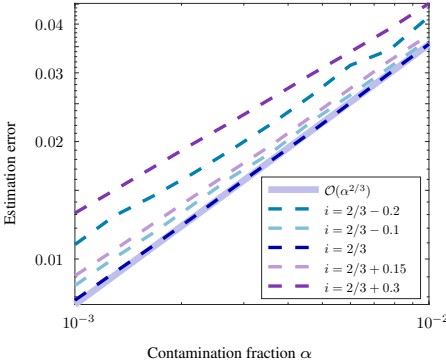

Figure 2: For any $k = \lceil 4\alpha^i n \rceil$, the error is at least $\mathcal{O}(\alpha^{2/3})$ for a sub-Gaussian distribution $\mathrm{p} \in \mathcal{P}_{\text{SG}}$.

Table 2: Parameter values of the numerical experiments.

| Figure | Theorem | Class | Distribution | $n$ | $\delta$ | $n_{\text{rep}}$ |
|--------|---------|-------|--------------|-----|----------|------------------|
| 1(a) | 3.2 | $\mathcal{P}_2$ | $\text{Pareto}(0.45; 1; 0)$ | $10^6$ | 0.05 | 100 |
| 1(b) | 3.3 | $\mathcal{P}_{1+r}$ | $\text{Pareto}(0.75; 1; 0)$ | $10^6$ | 0.05 | 100 |
| 1(c) | 5.1 | $\mathcal{P}_{\text{sym}}^{\varepsilon_0, c}$ | $t_3$ | $10^7$ | 0.05 | 100 |
| 2 | 4.2 | $\mathcal{P}_{\text{SG}}$ | Half-normal | $10^7$ | 0.05 | 100 |

**Finite Variance Distributions.** In Figure 1(a) we illustrate Theorem 3.2 by considering a Pareto distribution with finite variance. The figure shows that the empirical error of MoM aligns with the order of the upper bound from Theorem 3.2.

**Infinite Variance Distributions.** In Figure 1(b) we exemplify Theorem 3.3 using a Pareto distribution with infinite variance. Again, the figure shows that the empirical error of MoM aligns with the order of the upper bound from Theorem 3.3.

**Symmetric Distributions.** In Figure 1(c) we depict Theorem 5.1 using a Student's $t$ distribution. The figure shows that MoM continues to align with the corresponding theoretical upper bound in Theorem 5.1. Interestingly, the trimmed mean does not seem to exploit symmetry (in order) as effectively as MoM does. Although its error has not been characterized, the empirical results suggest that the trimmed mean estimator may be sub-optimal in these cases.

**Sub-Gaussian Distributions.** In Figure 2, we show that the order $\alpha^{2/3}$ cannot be improved for half-normal distributions for any choice for the number of blocks $k$. Specifically, we plot the error of MoM with different choices for the size of the blocks $k = \lceil 4\alpha^i n \rceil$. The dashed line with the steepest slope corresponds to $i = 2/3$ and the asymptotic bias $\alpha^{2/3}$, in accordance with Theorem 4.1 above.

