# OpenReview forum: "On the Optimality of the Median-of-Means Estimator under Adversarial Contamination"
_NeurIPS.cc/2025/Conference — NeurIPS 2025 poster_

### Official Review · Reviewer_QWU9 · 2025-06-30

**Clarity:** 2
**Significance:** 2
**Originality:** 3
**Rating:** 4
**Confidence:** 3

**Summary:**

In this paper, the authors present upper and lower bounds for the error of MoM under adversarial contamination for multiple classes of distributions. In particular, the authors show that MoM is (minimax) optimal in the class of distributions with finite variance, as well as in the class of distributions with infinite variance and finite absolute (1+r)-th moment. The authors also provide lower bounds for MoM’s error that match the order of the presented upper bounds, and show that MoM is sub-optimal for light-tailed distributions.

**Questions:**

Please check the points mentioned in "Strengths And Weaknesses.".

**Ethical Concerns:**

["NO or VERY MINOR ethics concerns only"]

**Final Justification:**

The authors have addressed my questions and comments. Given the discussions and rebuttal, I keep my score for acceptance.

**Limitations:**

Yes.

**Paper Formatting Concerns:**

No major formatting issues in this paper.

**Quality:**

3

**Strengths And Weaknesses:**

Strengths:

The paper correctly defines the problem. It explains the model, restrictions, related assumptions, and the main problem by addressing all related issues.

Weaknesses: The paper has some shortcomings, and I will mention them here:
1. In Definition 3.1, why do you use Q_m(q) to quantify the upper bound of estimation error? What is the insight behind the math?

2. In Theorem 3.1, could you please show more analyses for Theorem 3.1, since I cannot get more insights into this upper bound. For example, the performance with special cases could be discussed.

3. As for line 171, the authors say "Indeed, we assume only that the mean of the inliers is finite." I recommend that this assumption is given in an individual environment.

4. In Theorem 3.1, we can see that $\alpha<1/2$. In Theorem 3.2 and Theorem 3.3, why $\alpha\leq 0.4$ and $\gamma \in(2, 2.5]$? The assumption seems to be stronger.

5. In Lemma A.1, in line 453, the authors assume that the random variable follows binomial distribution, but I cannot see the variable q in the results.

6. In line 469, Why $\sqrt{}+\alpha m<1/2$ holds true? $m$ is the size of the block, for example, if $m=1000$, $\alpha=1/10$, $\alpha m=100$, which is not less than $1/2$.

In this upper bound, if we use Lemma A.1, why there exists $1/2+\alpha m$?

---

> ### Author Rebuttal · Authors · 2025-07-30
>
> We would like to thank the Reviewer for the constructive feedback provided. The following provides detailed answers and clarifications for the Reviewer’s questions. We are confident that the comments and questions raised are fully addressed below, and we would be happy to answer any further questions the Reviewer may have during the authors/reviewers discussion period.
>
> **Insights and intuition behind Theorem 3.1 and the usage of function $Q_m$**:
> The function $Q_m$ describes the quantile function of differences between averages of m i.i.d. samples and the actual mean. Since MoM carries out a median of averages corresponding to the samples in each block, its value is between two quantiles of averages of i.i.d samples (those corresponding to blocks without contaminated samples). The specific quantiles upper and lower bounding the value of MoM depend on the number of blocks without contaminated samples, which depend on the fraction of contaminated samples and on the number of blocks used, as described in the statement of Theorem 3.1.
>
>
> **Finite mean assumption for Theorem 3.1**: In the final version of the paper we will revise the sentence, in order to improve clarity, as follows:
> “In fact, no additional assumptions on the distribution of the inliers is required.”
> We emphasize that the mean has to be finite in order to be possible to estimate it.
>
> **Contamination tolerance in Theorems 3.2 and 3.3**: The contamination tolerance in Theorems 3.2 and 3.3 is lower than $1/2$ because of technical constraints in the proofs. This is a common phenomenon in robust mean estimation as many estimators have a contamination tolerance significantly lower than $1/2$. In fact, the results in the paper show that MoM tolerates more contamination than other estimators. Specifically, the results in the paper hold for all $\alpha\leq0.4$, whereas as described on lines 200-203, existing results establish a contamination tolerance of $\alpha\leq0.13$ for the trimmed mean, and of $\alpha\leq0.36$ for Catoni's M-estimator.
>
>
> **Binomial distribution in the proof of Lemma A.1**: There is not a variable $q$ in the results, $q$ is used as a dummy variable to explain the notation. In the final version, we will improve the clarity of the sentence in line 453 as follows:
> Where, for any $m\in \mathbb{N},q\in[0,1]$,  $\text{Bin}(m,q)$ denotes a random variable drawn from a binomial distribution of $m$ trials with a probability of success $q$.
>
> **Inequality in the proof of Theorem 3.1 (line 469)**: The inequality in line 469 is valid since $k>2\alpha n$. Specifically, since, $m=[n/k]$, the assumption $k>2\alpha n$ becomes  $ m < 1/(2\alpha)$. For instance, in the example mentioned by the Reviewer we have that $\alpha=1/10$ implies $m < 5$.
>
> In the proof of Theorem 3.1, $1/2+\alpha m < 1$ holds because for the choice of $k$ in the statement of the theorem (see equation (9)) we have that $k>2\alpha n$ so that $ \alpha m < 1/2$ since $m=[n/k]$.

---

> > ### Comment · Reviewer_QWU9 · 2025-08-01
> >
> > Thanks for the authors' responses. However, I still have a couple of questions which are listed below.
> >
> > 1. I think that the assumption on the mean of the inliers is required. Is this assumption used in the proofs? It is important for the results, so I would recommend to place the assumption in an individual environment.
> >
> > 2. As for the upper bound in Theorem 3.1, if $\alpha=0$, will the upper bound reduce to that in the attack-free case? On the other hand, if we have more samples, for example, we increase $m$, why will the upper bound be larger? I am sorry that I cannot understand. Please provide the analyses for each theorem, then readers could get insights into the results beyond math.

---

> > > ### Author Response · Authors · 2025-08-02
> > >
> > > We appreciate the Reviewer’s prompt feedback. The following provides detailed answers and clarifications for the Reviewer’s new questions.
> > >
> > > **Finite mean assumption**: The assumption about the finiteness of the mean was briefly mentionedin "Section 2 Preliminaries" (line 87) in the submitted manuscript. In the final version of the paper we will stress this fact modifying the sentence in line 171 as "Indeed, as mentioned in Section 2.1, the only assumption made about the underlying distribution is that its mean is finite."
> > >
> > > **Theorem 3.1 in the attack free case**: Firstly, Theorem 3.1 is a general result with no clear analogous in the attack-free case. For particular cases with analogous results in the attack-free case, the bounds in the paper reduce to those in the attack-free case. For instance, if $\alpha=0$, the bound in Theorem 3.2 (consequence of Theorem 1 for distributions with finite variance) reduces to that in the attack-free case because the term with $\alpha$ in bound (10) disappears so that such bound recovers the known sub-Gaussianity result for MoM in the i.i.d scenario (see also lines 192-194) in the submitted manuscript.
> > >
> > > **Upper bound when $m$ increases**: The upper bound does not increase when $m$ increases, the bound indeed decreases when $m$ increases.
> > > Specifically, the function $Q_m$ decreases towards zero when $m$ increases because it corresponds to the quantile function of a random variable $B_m$ (Def. 3.1) that has zero mean and variance of order $O(1/m)$. In addition, notice that the argument of $Q_m$ in the bound is bounded by $ 1/2 + \sqrt{m\log(2/\delta)/(2n)} + \alpha m $  and it does not increase with $m$. To see this, note that the second term does not increase with $m$ since $m\ll n$ and the third one does not either increase because $\alpha m \ll 1/2$.
> > >
> > > **Intuition behind theorems**: In the final version of the paper, we will make use of the extra space allowed to include the intuitive explanations described above together with others for the remaining theorems, as follows.
> > > All the upper bounds in the paper are special cases of Theorem 3.1. Specifically, in each theorem we bound the function $Q_m$ for a different class of distributions. In Theorem 3.2 we consider the class of distributions with finite variance. For these distributions, we have that $Q_m \leq O(1/\sqrt m)$. Therefore, as $k \sim \alpha n$ and $m=\lfloor n/k \rfloor$, we have that the error of MoM is upper bounded by $Q_m \leq O(\sqrt \alpha)$. In Theorem 3.3 we consider more general classes of distributions which contain distributions with heavy tails. In a heavy-tailed distribution, atypically large values are more likely to be drawn (the tails of the pdf decay slowly). Therefore, we obtain slower error rates than in Theorem 3.2. Specifically, we have that $Q_m \leq O(1/m^{\frac{r}{1+r}})$. For the theorems corresponding to cases with light tails, the results are obtained from Theorem 3.1 by characterizing the difference between the cdf function of $B_m$ and that of a Gaussian distribution  (e.g., using the Berry-Essen theorem).

---

> > > > ### Comment · Reviewer_QWU9 · 2025-08-05
> > > >
> > > > Thanks for reviewers' explanations. As for the second question, the response still cannot convince me. When we analyze the learning performance with regard to the upper bound of the learning error, especially for the adversarial scenario, we need to analyze the effects of each factor by fixing other parameters. Specifically, we study the upper bound in Theorem 3.1. Well, $m$ is the number of samples for each block. For example, take a look at the third term $\alpha m$. It is increasing with regard to $m$ since $\alpha>0$. As we know, the learning error will reduce as the number of samples increases. However, $\alpha m$ will increase when $m$ increase, which does not make sense. Although you said $\alpha m <\frac{1}{2}$, we only say the upper bound is not loose.

---

> > > > > ### Author Response · Authors · 2025-08-05
> > > > >
> > > > > We thank the Reviewer for the active engagement during the Authors/Reviewers discussion period.
> > > > >
> > > > > As the Reviewer points out, the argument of $Q_m$ in the upper bound of Theorem 3.1 increases with $m$. However, since $m<1/(2\alpha)$, the argument is bounded.  The bound in Theorem 3.1 decreases as m increases since the function $Q_m$ also depends on $m$ and converges to the zero function when $m$ increases (this is because $Q_m$ is the quantile function of a zero mean RV with variance $O(1/m))$. In the final version of the paper, we will clarify that "the reason the bound in Theorem 3.1 decreases with an increasing number $m$ of samples per block is that the function $Q_m$  converges to the zero function as $m$ increases."

---

> > > > > > ### Comment · Reviewer_QWU9 · 2025-08-06
> > > > > >
> > > > > > Thanks for authors' explanations for quantile function $Q_m$. There may be something wrong with my understanding for the insights behind the math. Could you please show me any relevant references which use quantile funcitons as the upper bound of the error? With my understanding, the argument of the quantile function is the probability when the random variable $B_m\le Q_m(\cdot)$. If the factor $m$ increases, then the probability will increase. This makes sense, since we have more samples for the evidence. However, from the definition of quantile function, this function is non-decreasing. Why does this function go to zero as we increase $m$? I am wondering if I was missing something.

---

> > > > > > > ### Author Response · Authors · 2025-08-06
> > > > > > >
> > > > > > > We are not aware of previous works that use quantile functions to obtain such type of upper bounds. The usage of such function to get the upper bounds for MoM is a novel contribution of the paper.
> > > > > > >
> > > > > > >
> > > > > > > As the Reviewer mentions, the argument $p$ of the quantile function is the probability with which the random variable $B_m$ is smaller or equal than $Q_m(p)$. Therefore, the value $Q_m(p)$ of the quantile function  satisfies $P( B_m\leq Q_m(p))=p$, that is, the value $Q_m(p)$ is the $p$-quantile of $B_m$. When $m$ increases, the random variable $B_m$ gets more concentrated around zero (it has zero mean and variance $O(1/m)$). Therefore, the quantiles of $B_m$ (i.e., the values taken by the function $Q_m$) get also more concentrated around zero. In the limit, $B_m$ will be the random variable that assigns probability 1 to zero so that all its quantiles are zero. For instance, as described on lines 180-182, the quantiles of a Gaussian random variable with zero mean and variance $1/m$ decrease towards zero at a rate $O(1/\sqrt m)$, e.g., the 90% percentile of such a Gaussian is approximately $1.28/\sqrt m$.

---

> > > > > > > > ### Comment · Reviewer_QWU9 · 2025-08-07
> > > > > > > >
> > > > > > > > Thanks again for detailed discussions. I do not have further questions, and I decide to keep my scores.

---

### Official Review · Reviewer_3535 · 2025-07-07

**Clarity:** 3
**Significance:** 2
**Originality:** 2
**Rating:** 2
**Confidence:** 5

**Summary:**

This paper studies the optimality of the Median-of-Means (MoM) estimator under adversarial contamination. It establishes error bounds for various distribution classes and shows that MoM is minimax optimal in settings with finite variance, certain heavy-tailed distributions, and symmetric distributions, while being sub-optimal for light-tailed cases. The theoretical results are supported by numerical experiments presented in the appendix.

**Questions:**

As aforementioned, could the authors provide further insights into why MoM underperforms in light-tailed settings, possibly suggesting modifications to address this gap?

**Ethical Concerns:**

["NO or VERY MINOR ethics concerns only"]

**Final Justification:**

I prefer to maintain my original score.

**Limitations:**

See 'Weakness' part.

**Paper Formatting Concerns:**

No Formatting Concern

**Quality:**

2

**Strengths And Weaknesses:**

Strengths
- The paper rigorously provides upper and lower bounds for the MoM estimator under adversarial contamination across various distribution classes.
- The paper not only proves MoM’s strengths but also highlights its sub-optimality for light-tailed distributions, offering a realistic assessment.

Weaknesses
- As the paper also illustrates, the number of blocks $k$ must be carefully tuned depending on the underlying data distribution to achieve optimal performance. However, the authors do not provide any practical guidance or theoretical insights on how to choose $k$ in practice. A more detailed analysis of how the block size affects the robustness and accuracy of the estimator would strengthen the work and make the results more actionable.
- The constant $C$ in the proposed error bounds is not tightly controlled or minimized. By providing a more detailed description of how to minimize this constant, or non-asymptotic results, could show the way to improve the practical applicability of the estimator.
- The paper identifies that MoM is sub-optimal for light-tailed distributions, but it doesn’t provide sufficient insight into why it struggles in these cases or potential strategies to improve performance under such conditions.
- While the paper presents theoretical results and bounds through solid mathematical proof, there is a lack of empirical validation or numerical experiments that demonstrate the practical effectiveness of the proposed bounds and estimators in real-world settings.
- The results are focused on specific classes of distributions (e.g., finite variance, heavy-tailed, symmetric), but they do not consider more complex distributions or non-i.i.d. settings. The assumptions about the underlying distribution may limit the applicability of the MoM estimator in diverse real-world data scenarios.

---

> ### Author Rebuttal · Authors · 2025-07-30
>
> We would like to thank the Reviewer for the feedback provided. However, we find some of the Reviewer’s comments unclear. The following provides detailed answers and clarifications for the Reviewer’s questions. We would be happy to answer any further questions the Reviewer may have during the authors/reviewers discussion period.
>
> **Underperformance of MoM for light-tails**: we appreciate this interesting comment by the Reviewer. As we point out in the paper, MoM’s performance improves when the distribution is light-tailed (Theorem 4.1). However, MoM does not achieve the optimal order since there exist light-tailed distributions whose median is significantly different from their mean. Therefore, as MoM underperforms for asymmetric distributions, it consequently also underperforms for some light-tailed distributions. This is because MoM estimates the mean as the median (biased estimator of the mean) of $k$ sample means (unbiased estimator). Therefore the bias of the median makes MoM underperform in distributions with asymmetry (median significantly different to the mean). That is, the error of MoM decreases slower for asymmetric distributions.
>
> Providing modifications for MoM to address this gap is beyond the scope of the paper, which is to analyze the optimality/suboptimality of MoM. One possible solution could be to use instead the Lee-Valiant estimator which is a modification of MoM that aims to correct the bias introduced by the median. However, the asymptotic bias of the Lee-Valiant estimator  has not been characterized for light-tailed distributions. It could be interesting to know whether Lee-Valiant’s estimator outperforms MoM or not.
>
> In the final version we will include this discussion after Theorem 4.2.
>
> **Generality of the considered distributions and non-i.i.d setting**:
> We do not really understand these comments from the Reviewer. The classes of distributions we consider are completely general and complex since we consider distributions with finite variance and even infinite variance. Moreover, the classes of distributions we consider are common in related work [23, 24].
>
> The setting considered in the paper of adversarial contamination is a general non-i.i.d. setting where an adversary replaces a fraction of the samples  with arbitrary values, and possibly creating dependences among the samples. Adversarial contamination generalizes many real-word situations such as additive contamination and Huber’s contamination model, among others.
>
> **Numerical experiments for a proposed estimator**:
> We do not really understand this comment from the Reviewer. We would like to stress that the paper does not propose an estimator.
> The goal of the paper is to theoretically analyze the optimality and suboptimality of the well-known MoM estimator for different classes of distributions. Therefore, practical applications are beyond the scope of this work. In fact, it is common in robust mean estimation papers to not include numerical experiments of real-world settings (e.g., [23, 8]).
>
> **Tuning of the number of blocks $k$**:  as we point out in lines 248-250, it is common for estimators to require a parameter (in the case of MoM, the number of blocks $k$) to be adjusted based on the underlying distribution. For instance, Catoni’s M-estimator requires selecting an appropriate function [24]. That said, the goal of the paper is to analyze the optimality/suboptimality of the MoM estimator for different classes of distributions. Therefore, finding a way to tune $k$ is beyond the scope of the paper.
>
> **Constant $C$ in the bounds**: As described above, the goal of the paper is to analyze the optimality/suboptimality of the MoM estimator for different classes of distributions. Therefore, the results in the paper aim to clarify the optimality of MoM and the constant used in the bound does not change the optimality of the estimator (see also lines 204-207 in the paper). This is the reason we did not minimized the constant, similarly as other works in the topic as [23].
> In Theorems 3.2 and 3.3 we include an asymptotic expression for the leading constant to give an intuitive idea of its value. In the proofs of Theorem 3.2 we give the exact value of the constant, which is
> $$
>   C = 2\sqrt{2+\sqrt{2}} \left( \frac{1}{(1/2-1/\gamma)^{3/2}} + \frac{\sqrt{\gamma}}{(1/2-1/\gamma)^{1/2}}   \right).
>    $$

---

### Official Review · Reviewer_SDfx · 2025-07-08

**Clarity:** 3
**Significance:** 2
**Originality:** 2
**Rating:** 3
**Confidence:** 4

**Summary:**

This paper studies the theoretical performance of the one-dimensional median of means estimator under adversarial contamination.

Over the class of distributions of finite $1+r$ absolute moment, for $r \in (0, 1]$, the paper proves new upper bounds on the $1-\delta$ quantile of the error of the median of means estimator (Theorem 3.2 and 3.3). These upper bounds, when combined with an exisiting minimax lower bound on this quantile, show that the median of means estimator is minimax optimal (up to a constant) over these classes of distributions. Previously, only the trimmed mean estimator was known to be minimax optimal in this sense.

The paper then considers light-tailed classes of distributions. Theorem 4.1 and 4.2 show that the performance of any median of means estimator, regardless of the choice of the number of blocks, deteriorates over the class of sub-exponential distributions and is no longer minimax optimal, with its error growing as $\alpha^{2/3}$ where $\alpha$ is the contamination level.

Finally, over certain special classes of symmetric distributions, the paper shows that the median of means estimator with a particular choice of number of blocks is minimax optimal.

**Questions:**

+ The theorems holds for a particular choice of number of blocks which depends on a unspecified constant $\gamma$ (and $\xi$ in Theorem 4.1). What is it?

+ In lines 275-276, it is claimed that the optimal error is of order $\alpha \log(1/\alpha)$ over sub-exponential classes. Can the authors provide a reference for this statement?

+ In Theorem 3.4: is the third moment assumption necessary? As far as I can tell, the statement is asymptotic (for sufficiently large n, no explicit estimate on n is given), and thus holds by the central limit theorem, so the second moment assumption should be enough.

+ In section 4.2, the class $P_{SG}^{s}$ is presented as subclass of subgaussian distributions. This class however is more closely related to gaussian distributions than subgaussian ones (equality of moments is required, rather than just a bound). I suggest modifying the notation and the language to make this clear.

**Ethical Concerns:**

["NO or VERY MINOR ethics concerns only"]

**Final Justification:**

My main concern with the paper was that the results are limited to one dimensional mean estimation. The authors promised to discuss this limitation in the writing which is a positive. Nevertheless, the comparable paper [1] - which studies the minimax optimality of the trimmed mean estimator under adversarial contamination - treats of the multivariate case, and on a technical level this is where most of the work is done.

The results of the current paper can still be of some interest, but I still think this is a borderline paper.

[1]: Lugosi, Gabor, and Shahar Mendelson. "Robust multivariate mean estimation: the optimality of trimmed mean." (2021): 393-410.

**Limitations:**

A main limitation of this work is the fact that it only deals with one dimensional mean estimation. This should be more prominently highlighted in the paper.

**Paper Formatting Concerns:**

I did not notice any major formatting issues.

**Quality:**

3

**Strengths And Weaknesses:**

Strengths:
+ The paper is well written and the results are properly contextualized.
+ The question of whether the median of means estimator is minimax optimal under adversarial contamination is interesting. The trimmed mean estimator is known to be robust to such contamination, but the situation, at least rigorously, is not as well understood for the median of means estimator.

Weaknesses:
+ The results are restricted to one dimensional mean estimation.
+ Though it is nice to have rigorous statements, the results are somewhat expected and may have limited impact on the research area.

---

> ### Author Rebuttal · Authors · 2025-07-30
>
> We would like to thank the Reviewer for the constructive feedback provided. The following provides detailed answers and clarifications for the Reviewer’s questions. We are confident that the comments and questions raised are fully addressed below, and we would be happy to answer any further questions the Reviewer may have during the authors/reviewers discussion period.
>
> **Dimensionality**: In this paper we have focused on the one-dimensional case since it is natural to study the MoM estimator only in this case. In higher dimensions, there is not a canonical extension of MoM since there is not a canonical (and efficient to compute) extension for the median. This is the reason related work for MoM has focused on the one-dimensional case [2-7].  Moreover, we  would like to emphasize that one-dimensional mean estimation is of high relevance, e.g., loss functions are univariate. One-dimensional focus is common in the robust mean estimation  literature beyond MoM estimator (e.g., [13, 24]). That said, we will describe this aspect of the paper more explicitly in the final version. Specifically, we will refer to MoM as a univariate mean estimator both in the abstract and in the introduction. Also, in Section 2.1 we will also emphasize that we are considering the one-dimensional mean estimation problem.
>
> **Impact of the results**: To the best of our knowledge the results presented in the paper have not been proved and stated in the existing literature. In fact, [24] explicitly states that "the asymptotic bias [of MoM] is not characterized and it is unclear whether the proposed estimator [MoM] achieves the minimax error bound." Theorem 3.2 in the submitted paper shows that MoM is minimax optimal under adversarial contamination for distributions with finite variance, while Theorem 3.3 extends the result to distributions with finite absolute $(1+r)$-th moment.
>
> Furthermore, as far as we know, Theorems 3.4 and 4.2 are the first lower bounds for the error of MoM under adversarial contamination. These lower bounds expose limitations of MoM that are new and unexpected based on existing results in the literature.
>
> Theorem 4.2 shows that MoM is suboptimal in the class of sub-exponential distributions. Notably, Theorem 4.1 together with Theorem 4.2 show upper and lower bounds with a matching order of $\alpha^{2/3}$ for the asymptotic bias of MoM in the class of sub-exponential distributions. Such suboptimal order for MoM was unknown and unexpected.
>
> **Meaning of the constant $\gamma$**: In Theorem 3.1, the constant $\gamma\in(2,2.5]$ can be thought as a dummy variable that describes the range of possible values for the number of blocks. More precisely, we have that the range of values for $k$ satisfying the result in Theorem 3.1 is
> $$k = \max \{ \lceil \frac{ \log(2/\delta)}{(1/2-1/\gamma)^2} \rceil, \lceil \gamma\alpha n \rceil \} $$
> for $\gamma\in(2,2.5]$.
>
> The above discussion also shows that  the optimality of MoM in Theorem 3.1  is valid not for just one particular $k$ but for all the range of possible values for $k$. This is desirable since it can give more flexibility in the choice of $k$ in practical scenarios. The same comment applies to the rest of the theorems as all upper bounds hold for ranges of values of $k$ described by a numeric constant.
>
> **Optimal error for sub-exponential distributions**: In lines 275-276 the paper mentions  that the optimal error is at most $\alpha\log(1/\alpha)$. This is because the error of the trimmed mean is at most $\alpha\log(1/\alpha)$ (to see this: apply the remark in page 10 from [23] to sub-exponential distributions).
>
> **Third moment assumption in Theorem 3.4**: As the reviewer points out, it is possible to have an asymptotic result by only requiring two finite moments. However, Theorem 3.4 is non-asymptotic, and that is why we need three finite absolute moments. Specifically, the result hinted by the reviewer holds by the CLT, which is asymptotic, whereas our result holds by Berry-Essen’s theorem, which is non-asymptotic. We appreciate the reviewer’s remark and in the final version of the paper we will comment that it is possible to have an asymptotic result under a second moment assumption.
> Furthermore, we will make Theorem 3.4’s statement more precise as follows:
>
> > Let $\widehat\mu$ be the MoM estimator with $k$ given as in (9) and (11). There exist positive constants $C,\alpha_\max,n_0$ and an adversarial attack such that for any $p\in\mathcal{P}_3,\alpha<\alpha_\max,n>n_0$,
> $$
> | \widehat\mu-\mu_p| \geq C \sigma_p \alpha^{2/3}
> $$
>  holds with probability at least $1-\delta$.
>
> **Naming of the class $\mathcal{P}^s_{\text{SG}}$**: We agree with the Reviewer that the class $\mathcal{P}^s_{\text{SG}}$ is closely related to Gaussian distributions. However, this class is contained in the class of sub-Gaussian distributions but not in the class of Gaussians. Therefore, we have chosen such notation and language because it fits better with the common thread of the paper (start analyzing the optimality or suboptimality of MoM in a class of distributions. Then, analize MoM’s optimality in a subclass.)

---

> > ### Comment · Reviewer_SDfx · 2025-08-02
> >
> > ***In higher dimensions, there is not a canonical extension of MoM since there is not a canonical (and efficient to compute) extension for the median.***
> >
> > The estimator in [1] is one such extension, and it achieves the minimax optimal rate of estimation under heavy tails.
> > It is the MoM analogue of the multivariate version of the trimmed-mean estimator studied in [2] - both are not easy to compute, but the goal here is to understand their statistical performance. I appreciate the fact that getting results in the multivariate case is challenging, but I do believe this to be an important limitation of the current work.
> >
> > I agree with the modifications to the paper the authors suggest to make this clearer.
> >
> > [1]: Lugosi, Gábor, and Shahar Mendelson. "Sub-Gaussian estimators of the mean of a random vector." (2019): 783-794.
> > [2]: Lugosi, Gabor, and Shahar Mendelson. "Robust multivariate mean estimation: the optimality of trimmed mean." (2021): 393-410.
> >
> > ***Meaning of the constant $\gamma$.***
> >
> > If you mean to say that the statement holds for ALL values of gamma between $2$ and $2.5$, then please make this clearer in the statements. In the current writing, it's not clear to me whether the statement says: for some $\gamma$ in this range, or for all $\gamma$ in this range.
> >
> > ***Optimal error for sub-exponential distributions.***
> >
> > Just because the trimmed mean achieves this rate does not mean that this is the minimax optimal rate - one needs a lower bound for that. I would suggest softening the claim in the paper.
> >
> > ***Third moment assumption in Theorem 3.4.***
> > As far as I can tell the revised statement still only requires the CLT. Unless you give an explicit estimate on $n_0$ - which likely depends on the third moment - It's not clear to me that a third moment assumption is necessary.

---

> > > ### Author Response · Authors · 2025-08-02
> > >
> > > We appreciate the Reviewer’s prompt feedback. The following provides detailed answers and clarifications for the Reviewer’s new questions. In case the Reviewer has some other follow up question or comment, we will be happy to provide further details during the discussion period.
> > >
> > > **Meaning of the constant $\gamma$**: In the final version of the paper we will make more clear that the theorem holds for all $\gamma\in(2,2.5]$. We will also update the other theorems that have a variable playing a similar role.
> > >
> > > **Optimal error for sub-exponential distributions**: As the Reviewer mentions, one needs a lower bound in order to characterize the optimal error.  Lines 275-276 in the paper do not aim to refer to an optimal error, just describe that "the optimal error is at most $\alpha\log(1/\alpha)$". Notice that such upper bound for the optimal error is sufficient to conclude that MoM is suboptimal for light-tailed distributions.
> > >
> > > We plan to update such a sentence as "the optimal error cannot be higher than $\alpha\log(1/\alpha)$ because the trimmed mean estimator achieves such an order (see remark in page 10 of [23])."
> > >
> > >
> > > **Third moment assumption in Theorem 3.4**: In our view, both results (the one with CLT or that with Berry-Essen) are of similar interest. It was not our intention to convey the idea that the finiteness of the third moment is essential for that type of result. In the final version of the paper we plan to state "We thank an anonymous reviewer for pointing out that the finiteness of the third moment is not essential for a result such as that in Theorem 3.4, and that a similar result can be obtained using the central limit theorem without requiring a finite third moment."

---

### Official Review · Reviewer_fw73 · 2025-07-17

**Clarity:** 4
**Significance:** 3
**Originality:** 3
**Rating:** 5
**Confidence:** 4

**Summary:**

This paper studies the asymptotic bias of the Median-of-Means (MoM) estimator for one-dimensional distributions under adversarial contamination, where a small fraction of the input samples can be arbitrarily corrupted. The authors present tight upper and lower bounds for the MoM estimator for several families of distributions.

**Questions:**

The authors are encouraged but not required to answer the questions mentioned in “Strengths & Weaknesses” above.

**Ethical Concerns:**

["NO or VERY MINOR ethics concerns only"]

**Final Justification:**

Most (if not all) of my comments are constructive suggestions. The authors responded that they would revise the paper accordingly. A few short and clear points that justify my rating were provided under "Strengths and Weaknesses."

**Limitations:**

yes

**Quality:**

4

**Strengths And Weaknesses:**

Strengths
+ The paper studies the problem of robust estimation under strong contamination models, which is of high relevance to the NeurIPS community.
+ The paper establishes several **tight** upper and lower bounds for the MoM estimator under adversarial contamination for different distribution families.
+ The paper is well written, and the presentation is clear.

Weaknesses
- The paper could be strengthened by highlighting its main technical contributions. For example, what are the paper's key ideas that are missing from prior work? Are there any known previous bounds for MoM estimators for the first four rows of Table 1?
- When defining the adversarial contamination model, one could further emphasize that the adversary can inspect the estimator/algorithm as well as the true mean. Related to this, on Line 114, the paper states: "there exists a distribution and an attack such that for any estimator...". The reviewer is a bit confused about the order of the quantifiers here -- why not "for any estimator, there exists a distribution and an attack..."?
- It could be helpful to clarify early in the paper that the authors focus on the one-dimensional case. The title can seem a bit of an overstatement as MoM is not optimal for all distributions. Perhaps "On the Optimality of..." or "When Is MoM Optimal under Adversarial Contamination?"
- The authors could consider citing [1].
[1] S. Minsker. Efficient Median of Means Estimator. COLT 2023.

Typos
- Line 121: should there be a comma after "as the number of samples increases (...)"?
- Lines 187 and 330: "there exist a positive constant"
- Line 205: "the bounds ... has".
- Line 318: "Gaussians distributions"
- Line 320: "infinite divisible"
- Line 342: "is specially well-suited"
- Line 438: "Propostion"

---

> ### Author Rebuttal · Authors · 2025-07-30
>
> We would like to thank the Reviewer for the constructive feedback provided and we are delighted that the Reviewer finds value in our work. The following provides detailed answers and clarifications for the Reviewer’s questions. We are confident that the comments and questions raised are fully addressed below, and we would be happy to answer any further questions the Reviewer may have during the authors/reviewers discussion period.
>
> **Main technical contribution**: In the final version of the paper we will be more clear explaining the main technical contribution of the paper. In particular, there are no known bounds for the first four rows of Table 1 prior to our work, as the asymptotic bias had not been characterized for these distribution classes.
> The error of estimators can be divided into two regimes: (1) for a reduced enough number of samples, the optimal error matches that in the i.i.d. scenario; (2) as the number of samples increases, the optimal error plateaus at a level determined by $\alpha$. The error in the second regime is known as the asymptotic bias and characterizes the optimality of estimators. While previous work on MoM [14] provides error bounds, there are four key aspects that are missing in their work that we solve in the paper:
>
> 1)  do not characterize the asymptotic bias.
>
> 2) results are restricted to the class of distributions with finite variance, and to the additive contamination model.
>
> 3) only provide upper bounds and not lower bounds.
>
> 4) guarantees hold only for reduced enough number of samples.
>
> In contrast, the results in our paper characterize the asymptotic bias of MoM for different classes of distributions. Moreover, we also provide lower bounds for the error.
>
>
> **Title**: We will revise the title to "On the Optimality of the Median-of-Means Estimator Under Adversarial Contamination" since it improves the paper as it accurately reflecets the paper’s contribution.
>
> **Reference and clarifications**: We appreciate the reference and we will include it in the final version.  We will also incorporate new comments that will improve clarification 1) on the order of the quantifiers on line 114 and 2) on the fact that our work focuses on the one-dimensional case. We are also thankful for the identified typos which will be fixed in the final version.

---

> > ### Comment · Reviewer_fw73 · 2025-08-07
> >
> > I thank the authors for their response and have no further questions at this point.
> >
> > I have read the discussions between other reviewers and the authors and have decided to retain my rating.

---

### Note · Authors · 2025-08-12

We would like to stress that the goal of the paper is to theoretically analyze MoM's optimality under adversarial contamination. MoM is a widely used one-dimensional estimator of the mean, so that identifying the conditions for its optimality and inherent limitations is essential for its reliable use.


The paper presents multiple key novel contributions:

1) Complete characterization of MoM's optimality across different classes of distributions. Prior work has been limited to small-sample regimes, lacks analysis of the asymptotic bias, focuses only on the family of distributions with finite variance, and considers the weaker model of additive contamination.

2) First lower bounds for MoM under adversarial contamination. These bounds show that, unlike other robust estimators, MoM cannot optimally exploit light-tailed distributions. This finding departs from prior expectations and reveals previously unknown weaknesses of the MoM estimator.

---

### Decision · Program_Chairs · 2025-09-17

**Decision:**

Accept (poster)

**Comment:**

This work studies the performance of the classical Median-of-Means (MoM) estimator for univariate distributions, including finite variance distributions, in the presence of a small constant fraction of arbitrarily corrupted datapoints. The main contribution is a collection of essentially tight upper and lower bounds on its performance for each of these families. With the exception of a highly problematic review, the reviewers overall appreciated the contribution. While the paper may be viewed as borderline by some (given the extensive amount of work in this area recently), I believe that the contribution is valuable and the submission is slightly above the acceptance threshold.